

# Subsurface iron accumulation and rapid aluminium removal in the Mediterranean following African dust deposition

Matthieu Bressac[1,2], Thibaut Wagener[3], Nathalie Leblond[4], Antonio Tovar-Sánchez[5], Céline Ridame[6], Samuel Albani[7,8], Sophie Guasco[3], Aurélie Dufour[3], Stéphanie H. M. Jacquet[3], François Dulac[8], Karine Desboeufs[9], and Cécile Guieu[1]

[1] Sorbonne Université, CNRS, Laboratoire d'Océanographie de Villefranche, LOV, Villefranche-sur-mer, 06230, France
[2] Institute for Marine and Antarctic Studies, University of Tasmania, Hobart, Tasmania, Australia
[3] Aix Marseille Univ., CNRS, IRD, Université de Toulon, MIO UMR 110, Marseille, 13288, France
[4] Sorbonne Université, CNRS, Institut de la Mer de Villefranche, IMEV, 06230 Villefranche-sur-Mer, France
[5] Department of Ecology and Coastal Management, Institute of Marine Sciences of Andalusia (ICMAN-CSIC), 07190 Puerto Real, Spain
[6] Sorbonne Université, LOCEAN, 4 Place Jussieu – 75252 Paris Cedex 05, France
[7] Department of Environmental and Earth Sciences, University of Milano–Bicocca, Milan, Italy
[8] Laboratoire des Sciences du Climat et de l'Environnement (LSCE), UMR 8212 CEA-CNRS-UVSQ, Institut Pierre-Simon Laplace, Université Paris-Saclay, 91191 Gif-sur-Yvette, France
[9] Laboratoire Interuniversitaire des Systèmes Atmosphériques (LISA), UMR7583 CNRS, Université de Paris, Université Paris-Est Créteil, Institut Pierre-Simon Laplace, France

*Correspondence to*: Matthieu Bressac (matthieu.bressac@imev-mer.fr)

**Abstract.** Mineral dust deposition is an important supply mechanism for trace elements in the low-latitude ocean. Our understanding of the controls of such inputs has been mostly built onto laboratory and surface ocean studies. The lack of direct observations and the tendency to focus on near surface waters prevent a comprehensive evaluation of the role of dust in oceanic biogeochemical cycles. In the frame of the PEACETIME project (ProcEss studies at the Air-sEa Interface after dust deposition in the MEditerranean sea), the responses of the aluminium (Al) and iron (Fe) cycles to two dust wet deposition events over the central and western Mediterranean Sea were investigated at a timescale of hours to days using a comprehensive dataset gathering dissolved and suspended particulate concentrations, along with sinking fluxes.

Dissolved Al (dAl) removal was dominant over dAl released from dust. Fe/Al ratio of suspended and sinking particles revealed that biogenic particles, and in particular diatoms, were key in accumulating and exporting Al relative to Fe. By combining these observations with published Al/Si ratios of diatoms, we show that adsorption onto biogenic particles, rather than active uptake, represents the main sink for dAl in Mediterranean waters. In contrast, systematic dissolved Fe (dFe) accumulation occurred in subsurface waters (~100-1000 m), while dFe input from dust was only transient in the surface mixed-layer. The rapid transfer of dust to depth (up to ~180 m d⁻¹), the Fe-binding ligand pool in excess to dFe in subsurface (while nearly-saturated in surface), and low scavenging rates in this particle-poor depth horizon are all important drivers of this subsurface dFe enrichment.

At the annual scale, this previously overlooked mechanism may represent an additional pathway of dFe supply for the surface ocean through diapycnal diffusion and vertical mixing. However, low subsurface dFe concentrations observed at the





basin scale (<0.5 nmol kg⁻¹) questions the residence time for this dust-derived subsurface reservoir, and hence its role as a supply mechanism for the surface ocean, stressing the need for further studies. Finally, these contrasting responses indicate that dAl is a poor tracer of dFe input in the Mediterranean Sea.

## 1 Introduction

Globally, iron (Fe) supply to the surface ocean sets the productivity of major phytoplankton groups (Moore et al., 2013). Among the multiple sources of Fe, atmospheric deposition of mineral dust represents an important supply mechanism in the low-latitude ocean (Duce et al., 1991; Jickells et al., 2005; Conway and John, 2014), and plays a key role in the functioning of Low Nutrient Low Chlorophyll (LNLC) systems (e.g., Guieu et al., 2014a). Despite widespread attention over the last three decades, large uncertainties remain in the factors controlling aerosol Fe solubility (Meskhidze et al., 2019).

Consequently, poorly constrained controls of Fe solubility partly explain the substantial inter-model difference in the atmospheric dFe input flux to the global ocean (~1-30 Gmol yr⁻¹; Tagliabue et al., 2016), and hinder accurate predictions of the impact of dust on ocean productivity.

African dust deposition events have long been known to impact trace element concentrations and fluxes in the upper water column of the Mediterranean (e.g., Buat-Ménard et al., 1988; Davies and Buat-Ménard, 1990; Quétel et al., 1993; Guerzoni

et al., 1999; Heimbürger et al., 2011). Our understanding of the role of dust in marine biogeochemical cycles remains limited, however, partly resulting from the difficulty in quantifying atmospheric dust fluxes to the surface ocean at short timescales. In the absence of direct assessments of atmospheric inputs, marine concentrations of tracers such as aluminium (Al) have been widely used to constrain these fluxes (e.g., Measures and Brown, 1996; Han et al., 2008; Anderson et al., 2016; Menzel Barraqueta et al., 2019). Al is predominantly of crustal origin and is characterized by a similar fractional

solubility than Fe with a longer residence time in seawater. Al could thus be used to constrain the integrated input of dust Fe over seasonal timescales (Dammshäuser et al., 2011). However, the fact that the distribution of Al can itself be controlled by the biological activity (e.g., Mackenzie et al., 1978; Middag et al., 2015; Rolison et al., 2015) questions its quality as a tracer. In addition, dust deposition being highly episodic in time and spatially patchy (Donaghay et al., 1991; Guieu et al., 2014a; Vincent et al., 2016), direct observations at sea are extremely challenging, and hence sparse (e.g., Croot et al., 2004;

Rijkenberg et al., 2008). To overcome this limitation, a variety of small-volume enclosed systems have been used to quantify Fe solubility from dust. Although yielding important insights into atmospheric trace element solubilities (Baker and Croot, 2010 and references therein), these systems do not fully simulate *in situ* conditions (de Leeuw et al., 2014), motivating the development of larger volume experiments (>100 L) where dust are free to sink and interact with dissolved and particulate organic matter while sinking (Bressac and Guieu, 2013; Guieu et al., 2014b; Herut et al., 2016; Gazeau et al., in revision).

Two key findings emerged from these large volume experiments. First, they allowed demonstrating the pivotal role played by the in situ biogeochemical conditions at the time of deposition in controlling post-depositional processes (i.e., dissolution, organic complexation, adsorption, colloidal aggregation), and their interplay (Wagener et al., 2010; Bressac and Guieu,



2013; Wuttig et al., 2013). The most striking and unexpected consequence is that upon deposition, dust can act as a net sink of dFe through scavenging (Wagener et al., 2010; Ye et al., 2011). Second, the large range in Fe solubility observed

depending on the season, reveals that oceanic rather than atmospheric conditions, determine in fine the flux of 'truly' bioavailable Fe to the surface ocean (Bressac and Guieu, 2013). However, these findings are valid in the first meters of the water column and direct observations of the whole water column are needed if we are to fully understand the role of dust in the oceanic iron cycle.

For this purpose, the Mediterranean Sea is a particularly relevant region. This semi-enclosed basin, characterized by a west-

to-east gradient in oligotrophy, receives some of the largest dust inputs of the ocean (Guerzoni et al., 1999), mostly under the form of wet deposition in the central and western part of the Basin, and a few intense events may account for the bulk of the annual deposition (Loÿe-Pilot and Martin, 1996; Vincent et al., 2016). The PEACETIME project (ProcEss studies at the Air-sEa Interface after dust deposition in the MEditerranean sea) and oceanographic campaign on board the R/V *Pourquoi Pas?* provided a unique opportunity to directly observe the biogeochemical effects of two mineral dust wet deposition events of

contrasted intensity that occurred during late spring 2017 in the central and western open Mediterranean Sea (Guieu et al., 2020). The presence of the R/V before, during, and/or few days after deposition allowed investigating (1) the parameters and processes shaping the contrasting distributions of dAl and dFe, (2) the importance of the timescale considered when assessing the flux of bioavailable Fe to the surface ocean, and (3) the relevance of using dAl to constrain dFe input from dust.

## 2 Materials and Methods

### 2.1 Oceanographic cruise

The PEACETIME cruise (doi.org/10.17600/15000900) was conducted during late spring conditions in May and June 2017 aboard the R/V *Pourquoi Pas?* in the central and western Mediterranean Sea. In total, 10 short stations (~8 hours) and 3 long stations located in the Tyrrhenian Sea (TYR; occupation = 4 days), the Ionian Sea (ION; 4 days), and in the western Algerian

basin (FAST; 5 days) were occupied (Fig. 1). FAST was an opportunistic station dedicated to investigate the biogeochemical effects of a dust deposition event by combining atmospheric and oceanographic in situ measurements before, during and after deposition (Guieu et al., 2020). At all stations, a 'classical' and a trace metal-clean (TMC) titanium rosette were deployed to sample the water column for biological and chemical parameters. Samples for aluminium and iron analyses were collected using the TMC titanium rosette mounted with GO-FLO bottles deployed on a Kevlar cable, while samples for

particulate Al (pAl) determination were also collected at all the stations from the classical rosette (see section 2.3).

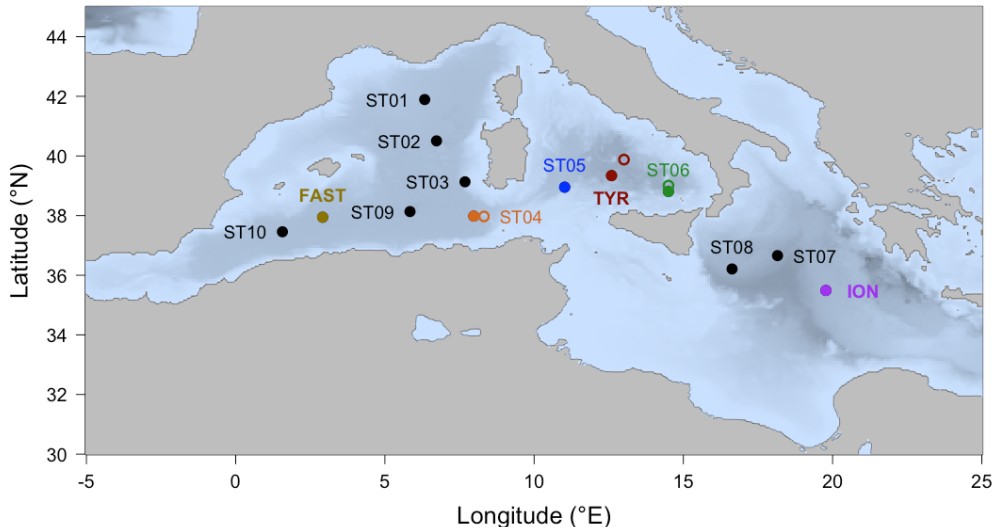

Figure 1: Sampling locations during the PEACETIME cruise (filled circles). The cruise track consisted of 10 short stations and 3
long stations (TYR, ION, and FAST). Open circles correspond to the stations 64PE370_12 (orange), 64PE374_13 (red), and
64PE374_12 (green) from the GEOTRACES GA04 section (Rolison et al., 2015; Gerringa et al., 2017) used for comparison in Fig.
2. Note that the same color code is used in figures 2, 5 and 6.

## 2.2 Dissolved Al and Fe concentrations

Immediately after recovery, the GO-FLO bottles were transferred inside a class-100 clean laboratory container. Seawater
samples were directly filtered from the GO-FLO bottles through acid-cleaned 0.2 µm capsule filters (Sartorius Sartobran-P-
capsule 0.45/0.2 µm). Dissolved Fe and Al samples were stored in acid-washed low-density polyethylene bottles and
immediately acidified to pH 1.8 (quartz-distilled HCl) under a laminar flow hood.

Dissolved Al analyses were conducted on board using the fluorometric method described by Hydes and Liss (1976). Briefly,
the samples were buffered to pH 5 with ammonium-acetate and the reagent lumogallion was added. The samples were then
heated to 80°C for 1.5 h to accelerate the complex formation. The fluorescence of the sample was measured with a Jasco FP-
2020 Plus spectrofluorometer (excitation wavelength 495 nm, emission wavelength 565 nm). Calibration was realized with
additions of Al standard solution in seawater. The detection limit (DL; 3 times the standard deviation (SD) of the
concentrations measured from the dAl-poor seawater used for calibration) varied between 0.2 and 0.5 nmol kg$^{-1}$. The reagent
blank determined by measuring acidified ultrapure water varied between 0.9 and 1.7 nmol kg$^{-1}$.

Dissolved Fe concentrations were measured (mostly on board in the class-100 clean laboratory) using an automated Flow
Injection Analysis (FIA) with online preconcentration and chemiluminescence detection (Bonnet and Guieu, 2006). The
stability of the analysis was assessed by analyzing daily an internal acidified seawater standard. On average, the DL was 15
pmol kg$^{-1}$ (3 times the SD of the concentration measured 5 times from the same dFe-poor seawater) and the accuracy of the
method was controlled by analysing on a regular basis the GEOTRACES seawater standards SAFe D1 (0.64 ±0.13 nmol kg$^{-1}$





($n$ = 19), consensus value 0.67 ±0.04 nmol kg$^{-1}$), GD (1.04 ±0.10 nmol kg$^{-1}$ ($n$ = 10), consensus value 1.00 ±0.10 nmol kg$^{-1}$)

and GSC (1.37 ±0.16 nmol kg$^{-1}$ ($n$ = 4), consensus value not available).

## 2.3 Suspended particulate trace elements

Just prior to sampling for particulate trace elements (pTM), GO-FLO bottles were gently mixed and pTM were sampled directly on-line from the pressurized (0.2 µm filtered N$_2$) GO-FLO bottles onto acid-cleaned 25 mm diameter Supor 0.45 µm polyethersulfone filters mounted on Swinnex polypropylene filter holders (Millipore), following GEOTRACES

recommendations. Filtration was stopped when the filter clogged or the bottle was empty. On average, each particulate concentration was obtained by filtering 4.8 L (range 1.1–10.2 L). When the filtration was complete, filter holders were transferred under a laminar flow hood and residual seawater was removed using a polypropylene syringe. Filters were stored in acid-cleaned petri-slides, left open under the laminar flow hood for ~24 h to allow the filters to dry. Particulate samples were digested (10% HF / 50% HNO3 (v/v)) following the protocol described in the 'GEOTRACES Cookbook' and

Planquette and Sherrell (2012). Procedural blanks consisted of unused acid-cleaned filters. Analyses were performed on a HR-ICP-MS (High Resolution Inductively Coupled Plasma Mass Spectrometry; Element XR, Thermo-Fisher Scientific). The accuracy of the measurements was established using the certified reference materials (CRM) MESS-4 and PACS-3 (marine sediments, National Research Council Canada) (Supp. Table 1).

In addition, pAl concentrations were also obtained at all the stations from the classical rosette. This additional pAl dataset

already published by Jacquet et al. (in revision) was obtained according to the sampling, processing and analysis methods described in Jacquet al. (2015). Briefly, 4 to 6 L of seawater collected with the Niskin bottles were filtered onto acid-cleaned 47 mm polycarbonate filters (0.4 µm porosity). Filters were rinsed with Milli-Q grade water and dried at 50°C. A total digestion of the membranes was performed using a tri-acid mixture (0.5 mL HF / 1.5 mL HNO3 / 1 mL HCl), and analyses were performed on the same HR-ICP-MS. A good agreement was obtained when comparing pAl concentrations obtained

with the TMC and classical rosettes at ION and FAST (difference in sampling time at TYR prevents quantitative comparison; see Sect. 4.1) (Supp. Fig. 2), demonstrating the absence of contamination for pAl when using the classical rosette.

## 2.4 Export fluxes and composition

Sinking particles were collected at ~200, 500, and 1000 m depth using PPS5 sediment traps (Technicap, France; 1 m$^2$

collection area) deployed on a free-drifting mooring for 4 (TYR and ION) and 5 days (FAST). Cups were filled with filtered seawater and buffered formaldehyde (2% final concentration) as a biocide. Once recovered, each cup representing 24 hours of collection was stored in the dark at 4°C until processed. Samples were treated following the standard protocol followed at the national service "Cellule Piège" of the French INSU-CNRS (Guieu et al., 2005) following the JGOFS' protocol. After removing the swimmers, the remaining sample was rinsed three times with ultrapure water in order to remove salt, and then





freeze-dried. The total amount of material collected was weighted to quantify the total exported flux. Several aliquots were then used to measure the following components: total and organic carbon, particulate Al and Fe, lithogenic and biogenic silica (LSi and BSi, respectively). Total carbon, particulate organic carbon (POC) (after removing inorganic carbon by acidification with HCl 2N), and particulate organic nitrogen (PON) were measured on an elemental analyzer CHN (2400 Series II CHNS/O Elemental Analyzer Perkin Elmer). For one sample (TYR 1000 m), 5 aliquots were analysed, yielding a

coefficient of variation (CV) of 6%. Particulate inorganic carbon (PIC) was quantified by subtracting POC from total particulate carbon. Particulate Fe and Al concentrations were determined by ICP-AES (Inductively Coupled Plasma Atomic Emission Spectrometry, Spectro ARCOS Ametek) after acid-digestion following the protocol described in Ternon et al. (2010). Blanks were negligible (<0.8% of the lowest Al and Fe concentrations of the digested aliquots) and the efficiency of the acid digestion was established using the CRM GBW-07313 (marine sediment, National Research Center for CRMs,

China) (Supp. Table 1). Samples for BSi and LSi (2 or 3 aliquots) were digested (NaOH at 95°C and HF at ambient temperature, respectively) and concentrations analysed by colorimetry (Analytikjena Specor 250 plus spectrophotometer) following the protocol described in Leblanc (2002). Mean export fluxes and composition of exported material are presented in Table 1.

**Table 1: Mean export fluxes for the mass, particulate organic carbon (POC), particulate inorganic carbon (PIC), particulate organic nitrogen (PON), biogenic silica (BSi), lithogenic silica (LSi), particulate aluminium and iron collected at TYR, ION and FAST (data not available at 500 m depth at TYR due to technical problems with the sediment trap). Values in parentheses correspond to the standard deviation of the arithmetic mean of the 4 (TYR and ION) and 5-day (FAST) deployment period.**

|  | Latitude | Longitude | Depth | Mass | POC | PIC | PON | BSi | LSi | Al | Fe |
|---|---|---|---|---|---|---|---|---|---|---|---|
|  | °N | °E | [m] | [mg m$^{-2}$ d$^{-1}$] | [µmol m$^{-2}$ d$^{-1}$] | [µmol m$^{-2}$ d$^{-1}$] | [µmol m$^{-2}$ d$^{-1}$] | [µmol m$^{-2}$ d$^{-1}$] | [µmol m$^{-2}$ d$^{-1}$] | [µmol m$^{-2}$ d$^{-1}$] | [µmol m$^{-2}$ d$^{-1}$] |
| TYR | 39.34 | 12.59 | 222 | 64.5 (54.3) | 340.7 (215.1) | 157.1 (109.1) | 40.2 (13.4) | 88.2 (78.7) | 339.6 (308.9) | 100.6 (101.9) | 25.3 (25.6) |
|  |  |  | 500 | - | - | - | - | - | - | - | - |
|  |  |  | 1056 | 76.0 (21.4) | 185.2 (49.1) | 225.2 (51.9) | 16.8 (4.5) | 86.0 (27.4) | 458.2 (192.1) | 135.6 (40.0) | 34.9 (11.6) |
| ION | 35.49 | 19.80 | 265 | 34.1 (19.1) | 315.6 (146.5) | 56.5 (29.9) | 31.3 (14.4) | 64.4 (53.1) | 126.8 (94.3) | 34.0 (28.3) | 8.5 (6.7) |
|  |  |  | 560 | 30.1 (10.2) | 144.2 (22.9) | 108.3 (37.2) | 12.6 (1.2) | 48.3 (15.6) | 125.7 (39.5) | 46.2 (18.3) | 11.9 (4.4) |
|  |  |  | 1097 | 30.1 (9.3) | 115.7 (16.5) | 98.7 (56.6) | 7.6 (2.0) | 36.3 (12.1) | 126.8 (36.5) | 45.7 (15.7) | 12.3 (3.9) |
| FAST | 37.95 | 2.91 | 259 | 29.1 (14.1) | 368.2 (139.5) | 67.6 (62.0) | 45.8 (20.6) | 90.8 (58.9) | 72.8 (48.6) | 26.5 (19.9) | 6.1 (4.5) |
|  |  |  | 498 | 33.9 (10.5) | 403.8 (150.3) | 116.6 (85.0) | 56.7 (26.0) | 60.7 ( 42.3) | 68.6 (64.3) | 27.1 (22.2) | 6.3 (5.2) |
|  |  |  | 1004 | 36.4 (17.7) | 379.0 (318.5) | 74.1 (44.2) | 53.0 (50.7) | 59.3 (29.8) | 82.4 (57.0) | 38.0 (19.6) | 9.1 (4.8) |





**2.5 Inventories and Kd**

Discrete measurements at different depths were used to calculate the water column integrated Al and Fe inventories (µmol m$^{-2}$) by trapezoidal integration. The concentration measured nearest to the surface was assumed to be constant up to 0 m. At FAST, six replicate measurements of dAl and dFe were performed at 5 and 400 m depth from 2 sets of 6 x GO-FLO bottles. The CV obtained at 5 and 400 m depths were used to determine the uncertainties in the 0-20 m and 0-200 m inventories,

respectively. Variability among replicates was higher for dFe (CV = 11.3 and 6.9% at 5 and 400 m depth, respectively) than for dAl (CV = 5.3 and 1.1% at 5 and 400 m depth, respectively), potentially reflecting a small scale variability in the dFe distribution.

At the FAST station, the partitioning coefficient between the particulate and dissolved phases (Kd; [particulate]/[dissolved]) was used to investigate exchanges between dissolved and particulate pools of Al and Fe. Following the relative change over

time of this metric allowed excluding potential artefacts related to change in water masses driven by lateral advection (Guieu et al., 2020).

**3 Results**

**3.1 Biogeochemical conditions**

The PEACETIME cruise took place in late spring when the stratification of the upper water column was well established

with the mixed-layer depth ranging between ~10 and 20 m along the cruise track. Chlorophyll *a* concentrations were typical of oligotrophic conditions (Guieu et al., 2020). A diatom-dominated deep chlorophyll maximum (DCM) that coincided with a maximum in biomass and primary production was well developed and observed all along the cruise track (Marañón et al., 2021). POC downward fluxes measured at 200 m depth were similar at the 3 long stations, while downward fluxes of Al and LSi, two proxies for dust, were maximum at TYR (Table 1). At the surface, dAl distribution was characterized by a marked

west-to-east increasing gradient (Supp. Fig. 1b) driven by advective mixing between (dAl poor) Atlantic and Mediterranean waters and by the accumulation of dust, and reflected by a strong relationship between surface dAl concentrations and salinity (Guerzoni et al., 1999; Rolison et al., 2015). All along the transect, dFe concentrations were high in the upper 100 m (up to 2.7 nmol kg$^{-1}$), and decreasing to levels <0.5 nmol kg$^{-1}$ below the euphotic layer (Supp. Fig. 1c). Subsurface patches of high dFe concentrations previously observed in the eastern Mediterranean Basin, and attributed to hydrothermal activity and

mud-volcanoes (Gerringa et al., 2017), were not observed along our cruise track.

**3.2 Dust deposition over the central and western Mediterranean Sea**

The impact in the water column of two dust deposition events of contrasting magnitudes could be studied during the cruise. They occurred in the area of the TYR and FAST stations (Fig. 1), on the 11-12 May and 3-5 June, respectively. The first





deposition event in the southern Tyrrhenian Sea was not directly observed but hypothesized based on satellite observations

of intense dust plume transport and water-column Al inventory presented in the following. The combined analysis of time-series of quick-looks of operational aerosol products from MSG/SEVIRI and from meteorological and dust transport models available during the campaign (Guieu et al., 2020) allowed us to suspect that a red rain event likely occurred over the southern Tyrrhenian Sea on the 11th of May and possibly on the early 12th, as illustrated in Supp. Fig. 3. The daytime daily mean aerosol optical depth (AOD) product over oceanic areas (Thieuleux et al., 2005) shows that a large dust plume was

exported from the Tunisian and Libyan coasts towards southern Italy and Greece from the 10th of May, with up to 1.5-1.6 AOD at 550 nm ($AOD_{550}$) (i.e., about 2 g m$^{-2}$ of dust in the column assuming a specific extinction cross-section of dust of 0.77 m$^2$ g$^{-1}$ following Dulac et al., 1992) on the 12th north of Sicily in the area of the Tyrrhenian stations reported in Fig. 1, a rather high value relatively unusual in this area (Gkikkas et al., 2016). On the 11th, clouds developed over most of the area affected by dust, whereas clear (turbid) sky was again present during daytime on the 12th (Supp. Fig. 3a). The extension and

dynamics of this dust transport event was reasonably well forecasted by the various existing regional dust transport models available during the campaign including the set of models of the World Meteorological Organization Sand and Dust Storm Warning Advisory and Assessment System operated by the Barcelona Supercomputing Center (BSC; Huneeus et al., 2016), the SKIRON/Dust model operated by the Atmospheric Modeling and Weather Forecasting Group of the National University of Athens (Spyrou et al., 2010), the CAMS model operated by the European Centre for Medium-range Weather Forecast

(ECMWF; Flemming et al., 2015), the NCEP/GFS model operated by the U.S. National Weather Service (Han et al., 2017), the NAAPS model modified from that of Christensen (1997) and operated by the U.S. Naval Research Laboratory, and the TAU/DREAM8 model operated by the Weather Research Centre of the Tel Aviv University (TAU; Kishcha et al., 2008). The dust plume extension in the cloudy area on the 11th is illustrated by Supp. Fig. 3b and c. Most meteorological models predicted significant precipitation over the Tyrrhenian Sea on the 11th (Supp. Fig. 3d), and until the morning of the 12th for

some of them (not shown). Dust transport models producing dust deposition fluxes generally forecasted dust wet deposition on that day between Tunisia and Italy, but with significant variability on the location, extent and schedule. The NMMB/BSC and SKIRON models predicted a significant wet deposition flux of dust, with up to 1.5 g m$^{-2}$ over 6 h, or more in the area of our stations 5, TYR, and 6 in the afternoon of the 11th of May (Supp. Fig. 3e and f). The DREAM model versions operated by the BSC and TAU, however, forecasted much lower values or even no dust wet deposition in the Tyrrhenian stations area

(Supp. Fig. 3g). For simplicity, the 11th of May 2017, 18:00 UTC will be considered as the time of deposition, that is approximately 3 to 10 days before our sampling of the area.

During the early June deposition event in the western Algerian basin, precipitation was directly observed in the area of the R/V and even sampled onboard (Desboeufs et al., in prep.), associated with a dust transport event of moderate extent and intensity over the southwestern Mediterranean basin. The $AOD_{550}$ peaked at about 0.40 in the area of the FAST station

(Desboeufs et al., in prep.), corresponding to a maximum columnar dust load <0.4 g m$^{-2}$, assuming a non-dust background $AOD_{550}$ in the boundary layer of 0.10-0.15 as observed north of the plume or the day before the plume arrived. This dust plume encountered a massive rain front covering ~80,000 km$^2$ and moving eastward from Spain and North Africa regions





(Desboeufs et al.., in prep.). Direct atmospheric and oceanographic observations of this event were possible thanks to a dedicated 'fast action' strategy (see Guieu et al. (2020) for details). Two rain periods concomitant with the dust plume
transported in altitude (1 to 4 km) allowed below-cloud deposition of dust in the FAST station area, as confirmed by on-board Lidar records (Desboeufs et al., in prep.). The first rain period occurred the 3$^{rd}$ of June in the neighbouring area of the R/V, and the second one occurred from the 4$^{th}$ (22:00 UTC) to the 5$^{th}$ of June (9:00 UTC), and was sampled on board the R/V the 5$^{th}$ of June from 00:36 to 01:04 UTC (Desboeufs et al., in prep.). This second rain event was characterized by a clear dust signature revealed by its chemical composition, representing a dust flux of about 40 mg m$^{-2}$ (Desboeufs et al., in prep.).
This sampled flux, considered as relatively modest compared to the multi-year record in this area (Vincent et al., 2016), was likely in the lower range of the total dust deposition flux that affected the whole area between the 3$^{rd}$ and 5$^{th}$ of June.

### 3.3 Reconstruction of the dust deposition fluxes

The absence of direct measurement of the dust deposition flux over the Tyrrhenian Sea and the limited spatial coverage of collection of atmospheric dust and rain at the FAST station call for an alternative approach to estimate dust deposition
fluxes. For this purpose, we used the water-column Al inventory. We acknowledge that this approach involves uncertainties, as all the observational approaches employed so far to quantify deposition (Anderson et al., 2016). Caveats include (1) other sources of pAl (which is unlikely in the open Mediterranean Sea during the surface water stratification period), and (2) some uncertainties into the derived dust fluxes that could come from the sampling method (Twining et al., 2015a), the time lag between deposition and sampling favouring dispersion of dust by lateral mixing, and to a lesser extent, the limited vertical
resolution below 500 m depth (Fig. 2a-d).

### 3.3.1 Central Mediterranean Sea

In the absence of pre-depositional observations and historic pAl data (to the best of our knowledge), the median pAl vertical profile obtained during the cruise at the other stations unimpacted by this event (grey bold line on Fig. 2a-d), similar or slightly higher than pAl data available for the open Mediterranean Sea (e.g., Sarthou and Jeandel, 2001), was used as a
background level. The comparison between the measured pAl vertical profiles and this background level revealed a marked excess in pAl (Al$_{excess}$) south of Sardinia (ST04) and in the southern Tyrrhenian (ST05, TYR and ST06; Fig. 2a-d and Table 2). This spatial extent is in good agreement with the maps of precipitation and dust wet deposition provided for the 11$^{th}$ of May by the ARPEGE, SKIRON, and NMMB/BSC models (Supp. Fig. 3).



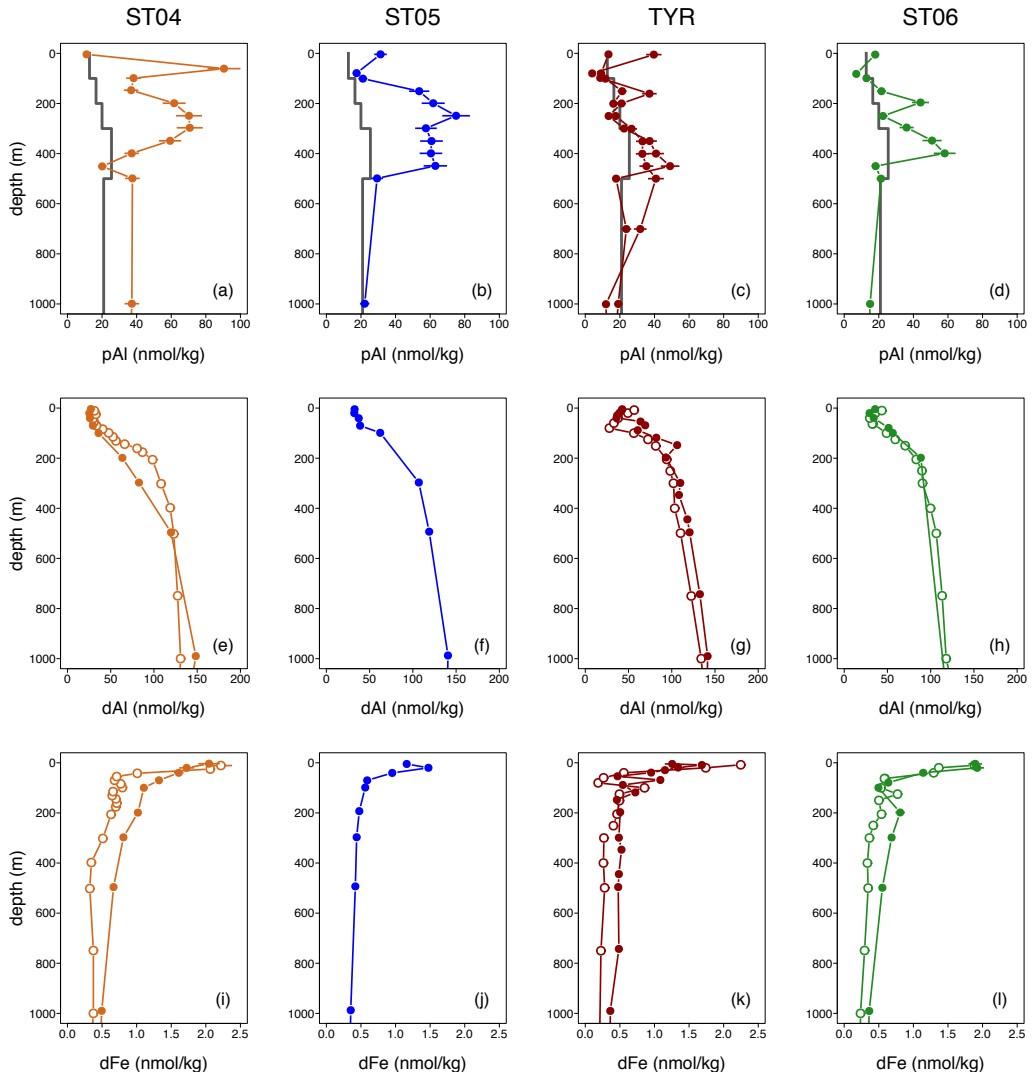


**Figure 2: Vertical distribution of pAl (upper panels), dAl (middle panels), and dFe concentrations (lower panels) obtained during the PEACETIME cruise (filled circles) at ST04 (a, e, i), ST05 (b, f, j), TYR (c, g, k), and ST06 (d, h, l). Particulate Al concentrations in (c) correspond to the vertical profiles TYR_2 and TYR_3 (Fig. 5). Previously published vertical profiles of dAl (Rolison et al., 2015) and dFe (Gerringa et al., 2017) obtained at similar locations (Fig. 1) are superimposed (open circles). The bold lines in a-d correspond to the median pAl vertical profile used as background level (see Sect. 3.3.1 for details).**

Assuming that Al represents 7.1% of the dust in mass (Guieu et al., 2002), a dust deposition flux ranging between 1.7 (ST06) and 8.9 g m$^{-2}$ (ST04) was derived from the Al$_{excess}$ inventory (Table 2). This range of dust deposition flux is of similar magnitude to the annual flux observed during former periods in the west central (7.4 g m$^{-2}$ yr$^{-1}$; Vincent et al., 2016) and northwestern Mediterranean Sea (11.4 g m$^{-2}$ yr$^{-1}$; Ternon et al., 2010), highlighting the remarkable magnitude of this event.





Furthermore, this comparison with annual fluxes confirms that the annual deposition of African dust in the Mediterranean region is generally driven by only a few intense events (Loÿe-Pilot and Martin, 1996; Guerzoni et al., 1999; Kubilay et al., 2000; Desboeufs et al., 2018). The strong spatial variability of these dust flux estimates, with a marked west-to-east gradient,

might result from the varying time lag between deposition and sampling of the water column at these different stations (Table 2), but also from the patchiness of the rainfalls associated with the rain front (Supp. Fig. 3). Indeed, Vincent et al. (2016) showed that high deposition events in the western Mediterranean are often limited spatially although the associated dust plumes may affect a large part of the basin. By assuming that the deposition was spatially homogeneous over the southern Tyrrhenian, an Al export flux of more than 4000 $\mu$mol m$^{-2}$ d$^{-1}$ is needed to explain the difference in the Al$_{excess}$ inventory observed between ST04, ST05 and TYR (i.e., ~3.6 to 8.4 days after deposition). This order-of-magnitude

difference with the Al export flux measured at TYR ~5 to 8 days after deposition (136 ±40 $\mu$mol m$^{-2}$ d$^{-1}$; Table 1) indicates that the observed spatial variability was primarily driven by the precipitation patchiness rather than related to a sampling bias.

**Table 2: Estimates of the input of pAl, dAl and dust south of Sardinia (ST04) and over the Tyrrhenian Sea (ST05, TYR and ST06)**

**based on the 0-1000 m Al$_{excess}$ inventories. The relatively low vertical resolution at TYR_1 precludes accurate estimates.**

|  | time since depositon [1] | Al$_{excess}$ [2] | Al loss [3] | pAl input [4] | dAl input [5] | dust flux [6] |
|---|---|---|---|---|---|---|
|  | [d] | [$\mu$mol m$^{-2}$] | [$\mu$mol m$^{-2}$] | [$\mu$mol m$^{-2}$] | [$\mu$mol m$^{-2}$] | [g m$^{-2}$] |
| ST04 | 3.6 | 23684 | 485 | 24169 | 363 | 8.9 |
| ST05 | 4.5 | 17773 | 610 | 18382 | 276 | 6.8 |
| TYR_1 | 5.6 | - | - | - | - | - |
| TYR_2 | 6.4 | 3591 | 874 | 4465 | 67 | 1.6 |
| TYR_3 | 8.4 | 4837 | 1143 | 5980 | 90 | 2.2 |
| ST06 | 10.5 | 3126 | 1423 | 4549 | 68 | 1.7 |

[1] 11/05/17 18:00 UTC is considered as the time of deposition

[2] Difference between the measured and median 0-1000 m pAl inventories (see section 3.3.1 for details)

[3] Estimates based on the downward Al flux (1000 m depth; TYR) and assuming a constant flux over time

[4] Corresponds to Al$_{excess}$ corrected for Al loss

[5] Estimates based on an Al fractional solubility of 1.5% (Wuttig et al. 2013)

[6] Estimates based on a Al composition of the dust of 7.1% (Guieu et al. 2002)

### 3.3.2 Western Mediterranean Sea

At the FAST station, dissolved and particulate Al and Fe concentrations were measured at high temporal and vertical resolutions before, during, and after the wet deposition of dust (Supp. Fig. 4). About 6 h after deposition, the total (dissolved





+ particulate) Al and Fe inventories within the upper 20 m increased by ~145 and 48 μmol m$^{-2}$, respectively (Fig. 3a and c). This increase in the 0-20 m inventories was consistent but higher than the atmospheric Al and Fe fluxes collected on the R/V (~98 and 25 μmol m$^{-2}$, respectively; Desboeufs et al., in prep.). Based on the increase in the 0-20 m total Al inventory and assuming 7.1% Al in the dust (Guieu et al., 2002), a total dust input of 55 mg m$^{-2}$ was derived. Although direct collection of atmospheric dust aerosols represents the most straightforward approach for quantifying the dust flux, it has only a limited

spatial coverage. At the opposite, the upper water-column inventory integrated most of the patchy rainfalls associated with this large rain front. This difference in time and space integrations is best illustrated by the ~70% increase in the 0-20 m pAl and pFe inventories observed the 4$^{th}$ of June (Fig. 3c), i.e., several hours before the rainfall collected onboard the R/V and probably associated with surrounding precipitations. It must be noted that the water-column approach is also subject to uncertainties and we cannot exclude an under-estimation of the deposition flux due to the rapid sinking of the largest dust

particles (e.g., Bressac et al., 2012). However, no evidence of these fast-sinking particles was found deeper in the water column (Fig. 3d), nor within the sediment traps (not shown).

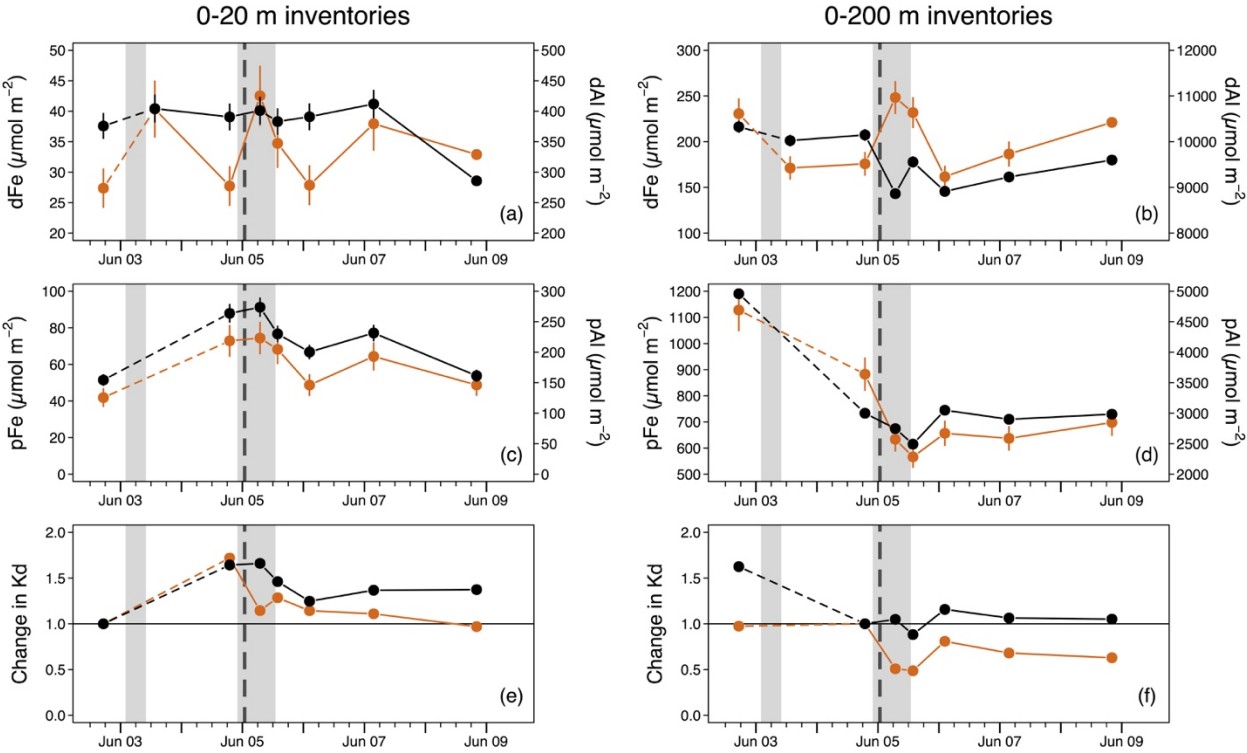

**Figure 3: Time evolution of the dissolved (a and b) and particulate (c and d) Al (black) and Fe (orange) inventories within the 0-20 m (left) and 0-200 m (right) depth ranges at the FAST station. Only dissolved inventories were measured the 3$^{rd}$ of June. The relative change in the coefficient of partitioning between the particulate and dissolved fractions (Kd) for Al and Fe is presented in e and f. Note that 0-200 m inventories measured the 4$^{th}$ of June (instead of the 2$^{nd}$ of June) were used for the initial Kd in f (see**



**Sect. 4.1 for details). Grey vertical bars indicate the two dusty rain events that occurred in the FAST station area. The grey-dotted**
**vertical line corresponds to the time of the dusty rainfall sampled on board the R/V (Desboeufs et al., in prep.).**

# 4 Discussion

## 4.1 Dust dynamic in the water column

In the Tyrrhenian Sea, deposition of dust was evidenced by the >3 times higher Al and LSi downward fluxes measured at 200 and 1000 m depth ~5 to 9 days after deposition relative to those measured at ION and FAST at the same depths (Table

1). At TYR, Al and LSi fluxes increased both by 35% between 200 and 1000 m depth, suggesting that a significant fraction of the dust particles was rapidly transferred to depth. This trend is consistent with the pAl vertical profiles at the 4 stations likely impacted by this event, as a subsurface maximum was depicted between ~200 and 500 m depth (Fig. 2a-d). In addition, three pAl vertical profiles performed at TYR over ~72 h showed a continuous decrease in surface pAl concentration of 20 $\mu mol\ m^{-2}\ d^{-1}$ that was accompanied by subsequent increases within the ~150-500 m depth layer (Fig. 4).

It is worth noting that ~5.6 days after the event, remarkably high pAl concentration observed at 1000 m depth (~260 nmol $kg^{-1}$ (TYR_1); not shown) could indicate that dust particles were sinking at a rate of ~180 m $d^{-1}$. This finding confirms that dust particles can be rapidly transferred to depth either alone (Bressac et al., 2012) or incorporated into biogenic aggregates (e.g., Hamm, 2002; Bressac et al., 2014; Laurenceau-Cornec et al., 2019; van der Jagt et al., 2018). Together, these observations demonstrate the atmospheric origin of pAl observed in the southern Tyrrhenian (rather than sediment

resuspension or advective inputs), and confirm that a significant fraction of the dust particles (coarse fraction) can rapidly leave the surface mixed-layer when the stratification is strong (Croot et al., 2004; Ternon et al., 2010; Nowald et al., 2015), while the remaining fraction (small-sized particles) likely accumulates along the thermocline until the disruption of the stratification (Migon et al., 2002).

At the FAST station, a two-orders of magnitude lower dust deposition flux (~55 mg $m^{-2}$) led to an increase by 78% of the 0-

20 m pAl inventory (Fig. 3c). About 24 h after deposition, only ~40% of this signal was still present in the mixed-layer. This is consistent with a short residence time in surface water for a significant fraction of the dust, although we cannot exclude the effect of lateral advection (Guieu et al., 2020). Deeper in the water column, the trend is more complicated to interpret with a 40% decrease (~2000 $\mu mol\ m^{-2}$) of the 0-200 m pAl inventory that occurred before/during deposition (Fig. 3d). This unexpected decrease cannot be explained by the vertical transfer of pAl, as only ~130 $\mu mol\ m^{-2}$ of pAl were exported out of

the upper 200 m over 5 days (data not shown). On the other hand, a southwestward flow disrupted the water column in the ~25-100 m depth range from the 3rd of June bringing water masses of distinct properties (Guieu et al., 2020). Therefore, it is likely that the water mass sampled before deposition (2nd of June) was different from the one sampled during the rest of the time-series. For this reason, inventories obtained the 4th of June (instead of the 2nd) were used as background level to investigate the temporal evolution of Kd(Al) and Kd(Fe) in the 0-200 m depth range (Fig. 3f).

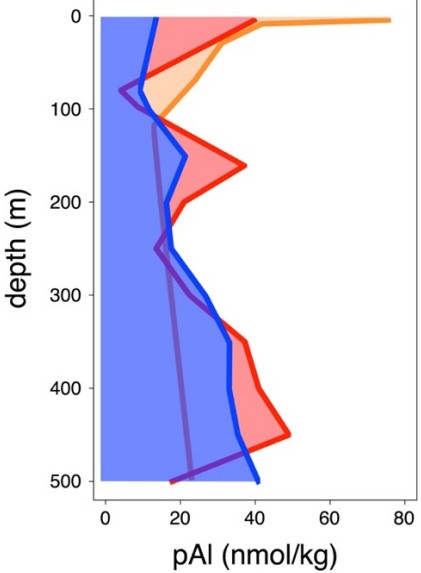


**Figure 4: Time evolution of the pAl vertical distribution measured at three different times over ~72 h at the TYR station. The orange (TYR_1), red (TYR_2), and blue (TYR_3) vertical profiles were measured approximately 5.6, 6.4, and 8.4 days after the dust deposition event, respectively.**

**4.2 Impact on the dAl inventory**

**4.2.1 Absence of dAl anomaly**

A relatively large range in Al fractional solubility (1-15%; defined as the fraction of dust-derived Al that dissolves in rainwater or seawater) has been reported (e.g., Orians and Bruland, 1986; Baker et al., 2006; Measures et al., 2010; Han et al., 2012). Assuming a conservative Al fractional solubility of 1.5% in seawater (Wuttig et al., 2013), dust deposition over the Tyrrhenian Sea led to a dAl input ranging between 68 and 363 µmol m$^{-2}$ (Table 2). Further assuming an homogeneous

distribution within the 0-20 m mixed-layer, this dust event represented a dAl input of 3.3-17.7 nmol kg$^{-1}$. However, the absence of noticeable anomaly in the long recognized relationship between surface dAl concentrations and salinity, when compared with published data (Rolison et al., 2015), reveals a limited or transient impact of this event on surface dAl concentrations (Fig. 5). Several mechanisms can be invoked here to explain the absence of dAl signal in the upper water column following the deposition. First, high surface dAl concentrations (>20 nmol kg$^{-1}$) might mask any additional input. At

ST06, the putative dAl input of 3.3 nmol kg$^{-1}$ (Table 2) is within the range of variability of both published and observed surface concentrations (30-43 nmol kg$^{-1}$; Fig. 2h). At the other end (ST04), the situation is different with a dAl input of 17.7 nmol kg$^{-1}$ that would represent more than 55% of pre-depositional surface dAl concentrations (31-32 nmol kg$^{-1}$; Rolison et al., 2015). Considering the short time lag between deposition and observations at ST04 (~3.6 days), and the very close to 1D dynamic condition in the TYR station area (A. Doglioli, pers. comm., 2020), it is unlikely that advective mixing diluted any





elevated dAl signal from this event. Deeper in the water column, no clear trend was obtained with subsurface dAl
       concentrations lower (ST04; Fig. 2e) or slightly higher than background levels (TYR; Fig. 2g). Similarly, no noticeable
       increase in dAl could be observed at the FAST station in the mixed-layer (Fig. 3a and 5). In contrast to Kd(Fe), Kd(Al) was
       still higher than pre-depositional value 4 days after deposition (Fig. 3e), potentially reflecting a lower fractional solubility for
       dust-derived Al relative to Fe, and/or higher removal rate for dAl. Below the mixed-layer, Kd(Al) remained relatively

constant and similar to initial value (Fig. 3f). Together, these observations indicate that wet deposition of dust over the FAST
       station area had a limited impact on the dAl inventory.

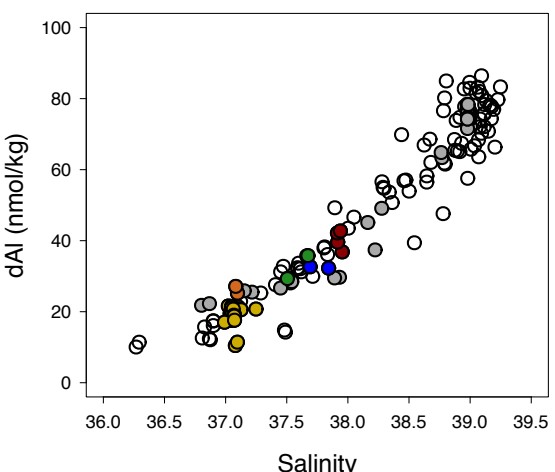

**Figure 5: Dissolved Al concentrations versus salinity measured in the upper 30 m of the Mediterranean Sea during the**
**PEACETIME (filled circles), 64PE370 and 64PE374 cruises (open circles; Rolison et al., 2015). Yellow, orange, blue, red, and**
       **green circles correspond to data obtained at FAST, ST04, ST05, TYR and ST06, respectively (Fig. 1).**

### 4.2.2 Drivers of the rapid removal of dAl

       An Al fractional solubility of 5% was measured in rainwater for dust aerosols collected at the FAST station (Desboeufs et
       al., in prep.), i.e., well above the conservative value of 1.5% used to estimate dAl inputs over the Tyrrhenian Sea. This
further supports the need of rapid dAl removal via adsorption and/or biological uptake to explain the absence of dAl
       anomaly following the dust events. In the Mediterranean Sea, a biological control on dAl distribution has been proposed to
       explain the strong coupling between dAl and orthosilicic acid ($Si(OH)_4$) in subsurface waters (Chou and Wollast, 1997;
       Rolison et al., 2015). In addition, several laboratory and field studies have demonstrated that marine phytoplanktons, in
       particular diatoms (mainly incorporated into the frustules; Gehlen et al., 2002), can uptake and/or scavenge dAl (Mackenzie
et al., 1978; Orians and Bruland, 1986; Moran and Moore, 1988; Loucaides et al., 2010; Twining et al., 2015b; Wuttig et al.,
       2013; Liu et al., 2019). To investigate the respective role of particle adsorption and biological uptake in removing dAl, Al
       was compared to Fe – a particle-reactive and bioactive element (Tagliabue et al., 2017) predominantly of crustal origin in the





Mediterranean Sea – through the Fe/Al content of suspended and sinking particles collected at different depth horizons (Fig. 6).

For suspended particles, the median Fe/Al ratio was maximum within the surface mixed-layer, and minimum at the DCM (60-100 m; Fig. 6a), highlighting a strong contrast in Fe/Al between the diatom-dominated particle assemblage at the DCM (Marañón et al., 2021) and detrital/lithogenic particles in the rest of the water column. This contrast supports the important role played by phytoplanktons, and in particular diatoms, in accumulating Al via active uptake (Gehlen et al., 2002; Liu et al., 2019) and/or adsorption onto cell membranes (Dammshäuser and Croot, 2012; Twining et al., 2015b). Regarding sinking

particles collected at ION and FAST, Fe/Al was strongly correlated with the relative proportion of LSi and BSi ($R^2 = 0.78$, p <0.001; Fig. 6b). Interestingly, this linear model predicts a Fe/Al ratio for BSi of ~0.22 mol mol$^{-1}$ (y-intercept), similar to the value observed in the diatom-dominated DCM (Fig. 6a). At TYR, the large dust input likely masked the signature of diatoms, as the median Fe/Al ratio in sinking particles (0.25 mol mol$^{-1}$) was similar to the Fe/Al ratio obtained for (1) suspended particles in the dust-impacted mixed-layer (Fig. 6a), and (2) particulate phase of the dusty rainwater sampled at

FAST (0.26 mol mol$^{-1}$; Desboeufs et al., in prep.). Sparse Al/Si ratios available for natural diatom communities range between ~1 and 10 µmol mol$^{-1}$ (van Bennekom et al., 1989; Gehlen et al., 2002; Koning et al., 2007). Using this range of values, the Al downward flux at 200 m depth driven by (and incorporated into) BSi would represent only 0.1-1.1 µmol m$^{-2}$ d$^{-1}$, i.e., ~0.2-3.3% of the total Al flux. This two-orders of magnitude difference with our conservative estimates of dAl inputs over the Tyrrhenian Sea (68-371 µmol m$^{-2}$; Table 2) indicates that adsorption onto biogenic particles (including BSi), rather

than active uptake by diatoms, was likely the main sink for dAl in that region.

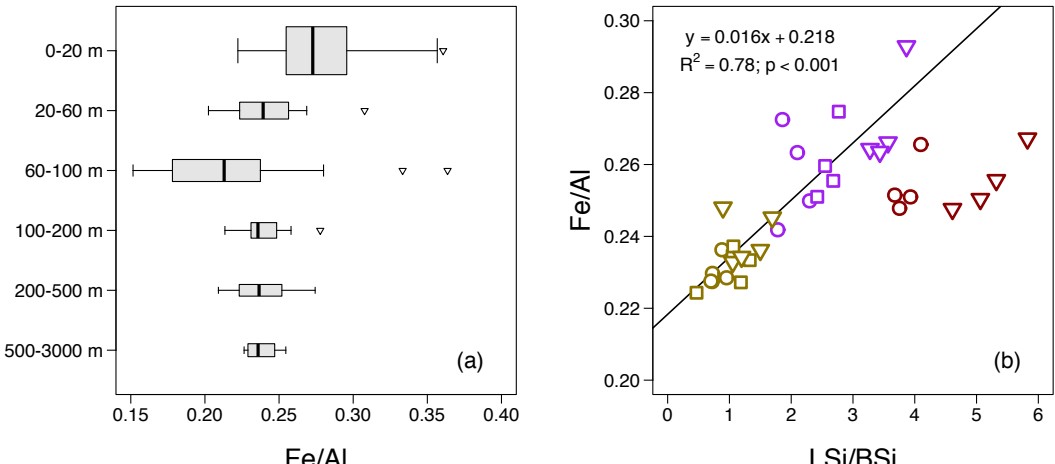

**Figure 6: (a) Box whisker plot of the Fe/Al molar ratio (mol mol$^{-1}$) for suspended particles collected at TYR, ION, and FAST. The**
**Fe/Al median values are 0.27 ($n = 37$; 0-20 m), 0.24 ($n = 13$; 20-60 m), 0.21 ($n = 17$; 60-100 m), 0.24 ($n = 12$; 100-200 m), 0.24 ($n = 9$; 200-500 m), and 0.24 ($n = 10$; 500-3000 m). For clarity, two outliers (Fe/Al = 0.50 and 1.02) observed in the 0-20 m depth range are**





**not represented. (b) Fe/Al versus LSi/BSi molar ratios (mol mol⁻¹) of sinking particles collected at ~200 m (circles), 500 m (squares), and 1000 m depth (triangles) at the stations TYR (red), ION (purple), and FAST (yellow). The black line corresponds to the best fit of a linear model (data obtained at TYR were excluded from this linear model).**

### 4.3 Vertical pattern in dFe input

#### 4.3.1 Transient dFe increase in the surface mixed-layer

The absence of pre-depositional observations in the Tyrrhenian Sea is more problematic for Fe compared to Al, as no clear longitudinal trend has been reported in the Mediterranean Sea for that element. Dissolved Fe vertical profiles were thus compared to previously published data that were obtained at similar locations (Fig. 1), and at the same period of the year for ST04 (mid-April) but about 2 months later at TYR and ST06 (early August) (Gerringa et al., 2017). Consequently, this approach ignores interannual and seasonal variabilities in dFe, and cannot be used to strictly quantify dFe input but remains valuable to investigate its magnitude and vertical distribution.

Assuming a Fe content of 4.45% in dust (Guieu et al., 2002), this dust event over the Tyrrhenian Sea represented a Fe input of ~1300-7000 µmol m⁻² (with a short retention time within the sea surface microlayer (Tovar-Sánchez et al., 2020)). Yet, dFe concentrations within the surface mixed-layer were at background levels (ST04 and ST06) or slightly below (TYR) (Fig. 2i-l). These observations made ~3 to 10 days after deposition indicate that this event had no impact on dFe in the surface mixed-layer at a timescale of days. At a shorter timescale, sampling performed at a high temporal resolution at the FAST station revealed two distinct increases of the 0-20 m dFe inventory that occurred during (+13 µmol m⁻²) and about 6 h after deposition (+15 µmol m⁻²; Fig. 3a). These ~50% increases were only transient and the pre-depositional level was rapidly recovered. Considering that Fe cycling in this LNLC system is dominated by physico-chemical rather than biological processes, our findings are consistent with a rapid scavenging of dFe in surface Mediterranean waters following dust deposition, as already reported in some mesocosm and minicosm dust addition experiments (Wagener et al., 2010; Wuttig et al., 2013; Bressac and Guieu, 2013). Overall, the Fe-binding ligand pool is nearly saturated in surface Mediterranean waters (Gerringa et al., 2017). As a consequence, any new input of dFe will tend to precipitate, pointing to the importance of the initial dFe and Fe-binding ligand concentrations in setting the net effect of dust input on dFe in the surface mixed-layer (Ye et al., 2011; Wagener et al., 2010; Wuttig et al., 2013).

#### 4.3.2 Enrichment in dFe below the surface mixed-layer

A key feature in the southern Tyrrhenian was the systematic subsurface excess in dFe observed from ~40 m (ST04) and 200 m depth (TYR and ST06) (Fig. 2i-l), and mirroring the vertical distribution of Al$_{excess}$ (Fig. 2a-d). Similarly, wet dust deposition over the FAST station area led to a net input of dFe mainly below the mixed-layer, as revealed by the opposite trends in Kd(Fe) observed in the 0-20 m and 0-200 m depth ranges (Fig. 3e-f). This increase in dFe relative to pFe was persistent on a timescale of days (Fig. 3f), and was primarily driven by dust dissolution (Fig. 3b) rather than ballasting of





pre-existing pFe (Fig. 3d), as evidenced by the low export Fe flux collected at 200 m depth (1.7-12.3 µmol m$^{-2}$ d$^{-1}$). This systematic excess in dFe observed below the mixed-layer and extending to 1000 m suggests that the mechanisms involved

are independent of the dust flux – that differed by two-orders of magnitude – and timescale considered (hours to week). Such dust-related subsurface enrichment in dFe (without enhanced surface dFe concentrations) has already been observed in the subarctic Pacific and tropical North Atlantic. This feature was attributed either to low oxygen levels allowing Fe(II) to stay in solution (Schallenberg et al., 2017), or to remineralization of organic matter formed in the dust-laden surface ocean (Measures et al., 2008; Fitzsimmons et al., 2013); two mechanisms that cannot be invoked here considering the oxygen

levels in subsurface (170-200 µM), the short timescale considered, and the low mesopelagic Fe regeneration efficiency (Bressac et al., 2019).

To account for this dFe excess below the surface mixed-layer, dust-bearing Fe must continue to dissolve as dust particles settle through the mixed-layer and reach the mesopelagic. The short residence time for dust in surface (Sect. 4.1), and the presence of a 'refractory' Fe pool within dust particles that dissolves over several days (Wagener et al., 2008) confirm that

dust dissolution can occur in subsurface. It is also likely that low particle concentration encountered at these depths relative to the particle-rich surface waters at the time of deposition prevented rapid removal of dFe (e.g., Spokes and Jickells, 1996; Bonnet and Guieu, 2004). Furthermore, the Fe-binding ligand pool is pivotal in setting the Fe fractional solubility (Rijkenberg et al., 2008; Wagener et al., 2008, 2010; Ye et al., 2011; Fishwick et al., 2014), and its magnitude, composition, and distribution likely shape patterns of dFe supply. While nearly saturated in surface, the Fe-binding ligand pool is in

relatively large excess to dFe in subsurface Mediterranean waters (Gerringa et al., 2017), and hence available to stabilize new dFe. Importantly, this subsurface pool is constantly replenished by bacterial degradation of sinking biogenic particles (Boyd et al., 2010; Velasquez et al., 2016; Bressac et al., 2019; Whitby et al., 2020). Thus, there is a permanent resetting of the ligand pool while dust particles settle (Bressac et al., 2019), and conceptually, we can imagine that the binding equilibrium between available ligands and Fe is rarely reached at these depths and timescale. This fundamental difference

with the surface waters (and batch experiments) could explain the high Fe fractional solubility of 4.6-13.5% derived in the southern Tyrrhenian from the increase in the 0-1000 m dFe inventories (relative to published data; Fig. 2i-l), and assuming 4.45% Fe in the dust (Guieu et al., 2002).

By feeding the subsurface dFe reservoir, dust deposition could represent an indirect supply route for the surface ocean through vertical mixing and diapycnal diffusion (e.g., Tagliabue et al., 2014). However, the residence time of this dust-

derived reservoir remains an open question. Relatively low subsurface dFe concentrations observed at the basin-scale (<0.5 nmol kg$^{-1}$; Supp. Fig. 1), compared to Atlantic waters for instance (Gerringa et al., 2017), argue in favour of a short residence time. Scavenging by sinking (dust) particles (e.g., Wagener et al., 2010; Bressac et al., 2019), and bacterial removal of humic-like ligands (Dulaquais et al., 2018; Whitby et al., 2020) represent two potential sinks for this subsurface dFe reservoir that need to be explored.



**5 Conclusions**

During the PEACETIME cruise performed in May-June 2017 in the western and central Mediterranean, the determination of the Al and Fe water-column distributions allowed us the observation at sea of two atmospheric wet deposition events, providing important insights into the timescale and pattern of dAl and dFe inputs from African dust in the remote Mediterranean Sea. The use of water-column Al inventory was needed – and successful – to assess dust deposition fluxes in
complement to atmospheric measurements and the 'fast-action' strategy used during the campaign to directly sample dusty rain events. Our observations show that dAl removal through adsorption onto biogenic particles was dominant over dAl released from dust at a timescale of hours to days. While surface dAl concentrations reflect seasonal changes and large scale patterns in dust deposition, this finding indicates that this tracer may not be appropriate to trace the imprint of a single dust deposition event in highly dust-impacted areas. Furthermore, dust deposition represented a significant input of dFe in the
surface mixed-layer only on a timescale of hours. On a longer timescale (days/weeks), dFe inputs occurred primarily below the surface mixed-layer and extended until 1000 m depth where the Fe-binding ligand pool likely in excess to dFe allows stabilizing any additional input of dFe. This mechanism may represent an additional pathway of dFe resupply for the surface ocean (through vertical mixing and diapycnal diffusion), although the residence time of this dust-derived dFe reservoir still needs to be investigated.

**Data availability**

Underlying research data are being used by researcher participants of the PEACETIME campaign to prepare other papers, and therefore data are not publicly accessible at the time of publication. Data will be accessible once the special issue is completed (June 2021) (http://www.obs-vlfr.fr/proof/php/PEACETIME/peacetime.php; last access 02/04/2021). The policy of the database is detailed here: http://www.obs-vlfr.fr/proof/dataconvention.php and
https://www.seanoe.org/data/00645/75747/ (last access 02/04/2021). In addition, dissolved Al and Fe datasets have been submitted for inclusion in the next GEOTRACES Intermediate Data Product (IDP) and will be publicly available when the IDP2021 is published.

**Author contribution**

M.B., T.W., and C.G. designed the study. M.B. wrote the manuscript. M.B., T.W., N.L., A.T.-S., C.R., S.A., S.G., and A.D.
collected and/or analysed the samples. F.D. and K.D. interpreted the atmospheric data. All the authors commented on and contributed to the improvement of the manuscript.



## Competing interests

The authors declare that they have no conflict of interest.

## Acknowledgements

We thank the captain and crew of the R/V *Pourquoi Pas?*, V. Tallandier for the deployment of the rosettes, the DT INSU for the design and preparation of the mooring line, N. Bhairy, G. De Liège, and G. Rougier for their help with the mooring. Hélène Ferré and the AERIS/SEDOO service are acknowledged for real-time collection during the cruise of maps produced from operational satellites and models used in this study, with appreciated contributions of EUMETSAT and ICARE for MSG/SEVIRI products, Météo-France for ARPEGE model outputs, the WMO SDS-WAS operated by the Barcelona

Supercomputing Center (BSC) for DREAM and NMMB model outputs, and the AM&WFG of the University of Athens for SKIRON model outputs. This study is a contribution to the PEACETIME project (http://peacetime-project.org; last access 02/04/2021), a joint initiative of the MERMEX and ChArMEx components supported by CNRS-INSU, IFREMER, CEA and Météo-France as part of the decadal programme MISTRALS coordinated by INSU. PEACETIME was endorsed as a process study by GEOTRACES and is also a contribution to IMBER and SOLAS international programs. M.B. was funded by a

Marie Sklodowska-Curie Postdoctoral European Fellowship (European Union Seventh Framework Programme (FP7/2007-2013) under grant agreement no. PIOF-GA-2012-626734 (IRON-IC project)). S.A. acknowledges funding from the European Union's Horizon 2020 research and innovation program under the Marie Sklodowska-Curie grant agreement 708119, for the project "DUSt, Climate, and the Carbon Cycle" (DUSC3).

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
