# Peer review of "Subsurface iron accumulation and rapid aluminium removal in the Mediterranean following African dust deposition"

_Biogeosciences, 2021_

## Referee Comment (RC2)

**Review bg-2021-87**

**Overview:**

This manuscript presents data on dissolved and particulates phases for iron and aluminium collected in the Mediterranean during the PEACETIME project. During the expedition a number of dust/precipitation events occurred and the authors attempt to derive dust fluxes from measurements before and after such events. Unfortunately the are missing some before data and have used data from other stations to try to bridge this gap, without taking into account possible differences in hydrology and past depositional histories between the stations concerned, this results in what appears to be a major overestimation of the estimated dust fluxes at some stations due to the use of inappropriate initial conditions. The lack of a Lagrangian hydrographic context for the stations also suggests that horizontal variations are being interpreted as temporal changes in the vertical.

**General comments:**

**Influence of Hydrography, Eulerian or Lagrangian approaches:**

A critical omission at present from this work is that there is no information on the hydrography and what is going on in the water. For a process study this is a major flaw as all too often changes in water masses may be interpreted falsely as the effect of rapid in water column processing. This is the reason why Lagrangian experiments are carried out in the ocean (Abbott et al., 1990; Boyd et al., 2005; Jickells et al., 2008; King et al., 2012; Krom et al., 2005; Law et al., 2005; Law et al., 2001) in order to remove the effects of water mass movement as much as possible. While the optimal way to carry out such a study is to use a tracer such as SF6 (Law et al., 2001), it can also be as simple as putting in a drifting buoy as the sampling point (King et al., 2012). It appears from the description in the associated paper (Guieu et al., 2020) that the same eulerian point was occupied for each station and no attempt at a quasi Langranian study was made, though that work mentions that a Lagrangian adaptive sampling strategy was employed, but this apparently refers to using satellite data to avoid placing stations at fronts or in eddies.

**Absence of a hydrographical context:**

I am really surprised that the authors did not think to use salinity as a tracer of recent precipitation events as this would also allow an estimate of the amount of rain that has fallen if there is minimal horizontal advection of water masses. It also appears that there is data for transmission and a LISST-Deep, both of which presumably would have picked up changes in the particles in the deep waters where the apparent excess particulate aluminium was located. Referencing data to density levels and examining the different water masses present (Schroeder et al., 2020) would help to resolve some of the issues around the lack of any hydrographical context in a future revision.

**Dissolved Al removal dominant over release from dust:**

Using the median profile as the initial starting point for looking at changes in the particulate concentrations has a number of apparent flaws that are not fully explained or investigated in the current manuscript. (i) There may be significant differences in the hydrography between the station used as the median profile and the experimental station (ii) similarly the experimental station may include a transient nepheloid layer due to water mass differences and lastly this may lead then to (iii) the time scales of the inferred changes being different, that is a seasonal or slow moving

transient change is interpreted as an instantaneous response to a dust event. In the present work section 3.3.1 at present does not give the reader a good idea of how the fluxes were calculated and what assumptions were made in the calculations. It appears that the difference between the median profile and the most recent profile is assumed to represent the total flux of particulate aluminium to the region in a single event, this seems very optimistic.

*Elevated particulate aluminium at depth but what about iron?* One of the main mysteries in this paper is if there is also a vast excess of particulate iron at stations ST04 and ST05 as there is for the particulate aluminium as this data is not provided (or at least I could not find it). It would be useful to include the particulate Fe data in Table 2 for instance as a check on the dust deposition estimates using Aluminium.

*Time scales*: Earlier work (Caschetto and Wollast, 1979) showed strong seasonal changes in dissolved aluminium concentrations, which would suggest that the residence time for dissolved aluminium is on the order of months and certainly less than a year for the Mediterranean. At present the manuscript basically interprets every change on the same timescale as between repeat measurements but care has to be taken when the sampling is not truly Lagrangian as you may confuse variability at depth with fast removal or dissolution processes. What the data in this paper do inform about however is the effect of deposition in surface waters and reconciliation of the timescales involved that control the strong salinity relationship with dissolved aluminium (Rolison et al., 2015) as it fits well to a two end member situation with low Al Atlantic surface water and high Al Eastern Mediterranean waters, however this implies rapid mixing of surface waters in the Mediterranean to maintain the observed gradient. Thus it would be helpful to examine in the current work how horizontal mixing may dilute and transport dissolved aluminium after a dust event. I appreciate that in the current manuscript this aspect is downplayed so as to simplify flux calculations but it does appear that it is a critical aspect for maintaining dissolved aluminium concentrations in the Mediterranean.

Work on thorium isotopes in the Mediterranean (Roy-Barman et al., 2002; Roy-Barman et al., 2009) also suggests that the dissolution of continental particles stored in the margins needs to also be considered. This does not appear to have been examined in detail by subsequent researchers but could be explored here briefly. Similarly for the elevated particulate aluminium it would be useful to explore this with reference to earlier work on the pressure effects on aluminium speciation (Moore and Millward, 1984).

*Role of colloids missing from this work:* A further obvious omission in this paper is any mention of the role of colloids for Fe (Bergquist et al., 2007) and Al (Dammshäuser and Croot, 2012; Moran and Moore, 1989) in the surface ocean. Given that colloidal Al could also be part of the measured particulate aluminium pool observed in this study it would also be of benefit to the reader if there was some discussion on the potential role of colloids in controlling the fractional solubility of Al in seawater (Dammshäuser and Croot, 2012; Dammshäuser et al., 2011; Dammshäuser et al., 2013; Moran and Moore, 1989).

*Biogenic silica in Saharan Dust*: Saharan dust can contain a significant amount of biogenic silica (Folger et al., 1967), either as phytoliths or freshwater diatoms. So there is the possibility that the biogenic silica may not be formed in situ. Alternatively the biogenic silica may be overestimated during the analysis due to lithogenic dissolution (Ragueneau et al., 2005). This typically is estimated using dissolved Al as a tracer for the lithogenic fraction, which appears not to have been done in this case. This is an opportunity missed really as a sequential dissolution protocol looking at both the Si and Al would have provided strong evidence for Al weakly bound to the particles (i.e. that released

with an acetic acid step as was done in other studies (Berger et al., 2008; Landing and Bruland, 1987; Planquette et al., 2009). At present it looks like the biogenic Si is overestimated.

*Precipitation estimates:* There are satellite data available for precipitation https://gpm.nasa.gov/data - did the authors examine this dataset to examine the spatial and temporal scales of precipitation during the expedition? This would likely support the model runs used currently. There are also rain rate data included in the Guieu et al. (2020) work associated with this expedition.

*Error estimates*: In the dust flux estimates there is no inclusion of error estimates and it would be useful to include these to constrain the values accordingly.

**Specific comments:**

- Line 57: In this context the work of Li et al. (2013) along with Moran and Moore (1988) are good examples of the apparent biological drawdown of Al.
- Line 68: It is not that unexpected as the deposition of dust is known to scavenge other elements from the water column also.
- Line 69: Do you mean fractional solubility with regard to aerosol solubility? As iron solubility in the water is a different measurement (Baker and Croot, 2010).
- Line 94: Please clarify that this is a Kevlar cable with a conducting insert in it and include information on the type of CTD that is on the rosette and the parameters measured on it.
- Line 95: The classical rosette is an aluminium oxide coated rosette? Please provide more information on this as the data in Supp Fig 2, do suggest that it is non-contaminating but the same data also suggest that the trace metal clean rosette could be contaminating (two obvious fliers). Alternatively it suggests there were major changes in the intermediate waters over a few hours (less likely).
- Line 95: Were nutrients taken from both sampling systems? There is no information here on if nutrients were taken and if so, how they were analysed.
- Line 112: There are no GEOTRACES consensus samples reported here for Al. Does this mean none were run? What data do you have that this method was giving good values?
- Line 113: Are the data corrected for the reagent blank?
- Line 120: The GSC consensus value is 1.54 ± 0.12 nM for Fe and the data are available here: https://www4.obs-mip.fr/wp-content-

omp/uploads/sites/31/2020/03/2019\_Consensus\_Values\_2009\_samples.pdf

- Line 125: The GEOTRACES Cookbook is citable (Cutter et al., 2010).
- Line 140: There are two or more fliers in Supp Fig 2, all of which come from the trace metal clean rosette so what is the reason for this? Some are in the surface so this is understandable but at 400 m at ION and 2000 m FAST, this seems more like a bad bottle as it would not be expected that the water masses would change the much in the few hours between casts. Are there any other elements that can help with determining if it is contamination from a bottle?
- Line 160: The usual description for this is that it is a sequential leach using the method of Paasche (1980) as modified by Nelson et al. (1989) using the spectrophotometric method of Mullin and Riley (1955). I would avoid using the thesis as a citation as it quotes the wrong paper for the silicate method and while it has the authors correct for silicate, the journal reference is actually a paper on phosphate (Murphy and Riley, 1962).

- Line 165: Table 1: The data for BSi, LSi, Al and Fe have high variability, with apparently one standard deviation being almost the same as the mean more so in many cases than the variability in the mass fluxes. This is therefore an inherently noisy data set, which likely represents the challenge of short term deployments, so it is important that this is also discussed in the manuscript in more detail as concentrating on the average fluxes presents a somewhat distorted picture.
- Line 172: Was this always in the mixed layer as observed by the hydrographical measurements. See the general comment above regarding the absence of any hydrographic context in this work.
- Line 185: How was the mixed layer defined here? A density or temperature criteria or something else? More details should be supplied.
- Line 219: As mentioned in the general comments above, there is satellite data for precipitation so there is no need to invoke model predictions. The authors could also use the change in surface salinity as an indicator.
- Line 239: Can the authors provide a little more information on how this dust flux was determined was it based on direct measurements of dust in the rain or via a proxy? This is important as without being able to see the Desbouefs et al. paper is may mean that this is actual a circular argument (e.g. that paper uses dissolved Al in the water to estimate the dust flux).
- Line 252: Using the median pAI profile without providing a hydrological context seems flawed in that it ignores differences in the hydrography. Additionally we are not provided with any statistical oversight of what the median profile represents. See the general comment on this above also.
- Line 256: The high particulate concentrations at depth may also represent a nepheloid layer. Is there any transmission data from the CTD to examine this possibility? The Guieu et al. (2020) work indicates that there was a transmissometer and a LISST sensor onboard that could answer these questions so it would be a good idea to include this data here. I am aware of earlier work which has shown that this area can form nepheloid layers at intermediate depths (Misic et al., 2008). Nepheloid layers have also been shown to be important for scavenging of Th and Pa in the Mediterranean (Gdaniec et al., 2018). Alternatively it could be from a mesoscale eddy so it would be good to rule that also using sea surface altimetry.
- Line 267: the units here are g m-2 but no time frame is given, this is then compared to an annual flux in g m-2 y-1 so how is the reader meant to compare this when we are not told how the deposition flux is calculated? The key missing piece of information is the assumed residence time for particulate aluminium and it seems here that the assumption (unstated) is that the excess particulate aluminium comes from a single dust event. See the general comments on this above.
- Line 270: This statement is confusing, as the inference appears to be that the excess dissolved Al all arrived in a single event? It is also not clear then how this relates to the rain event where the dust flux is estimated at 40 mg m-2.
- Lines 277 & 281: These two sentences are opposing each other and I don't follow here how if the deposition is assumed to be homogeneous that then the spatial variability indicates precipitation patchiness and not sampling bias. What type of sampling bias is inferred here?
- Line 285: Table 2 deposition not 'depositon'
- Line 285: Item 4: Why is the loss correction made? Particulate Aluminium in the traps is still in the water column at the time it is collected by the traps so it seems to be some sort of double accounting.
- Line 285: Item 6: The authors should check these numbers as using 7.1% and a molecular weight for Al of 26.98 gives a dust flux of 9.18 g m-2 for the ST04 station when using 24169  $\mu$ mol m-2 as the pAl input. Similarly 6.98 g m-2 for ST05.

- Line 289: The vertical profiles for the particulate Fe data are not shown in this work, only the integrated values and the depth/time when sampled.
- Line 298: This is perhaps a good example of why a Lagrangian framework is better to use than a eulerian one as clearly different water masses with different deposition histories are passing through the same site so the interpretation here based on a eulerian approach is flawed. See the general comment on this above.
- Line 300: So what is the excess Al seen at ST04 and ST05, slow sinking dust particles? Why then are they not seen at FAST?
- Line 305: As Kd is a ratio, using a normalization approach as shown in Figure 3 makes it harder to follow. Furthermore the use of the 4th of June for the 0-200 m inventory makes comparison with the 0-20 m data even more problematic Perhaps plotting the data as percentage of total Al might then be a better way to show how the relative amounts changed in the particulate and dissolved pools.
- Line 337: See the general comment about the lack of details on the hydrography.
- Line 340: Please include error bars in Figure 4 as it is not clear what the variability of the measurements is relative to the apparent changes seen here.
- Line 347: Another good reference for this which includes data at high dust loadings is Shelley et al. (2018) and in that work it is clear that the fractional solubility is very low under high dust loadings. Though it may be higher when deposition is as rain.
- Line 359: What is meant by '1D dynamic condition' here? That there is no lateral mixing? If the deposition is so patchy initially then of course lateral mixing will smooth this out quite quickly. Additionally horizontal mixing is typically orders of magnitude more than vertical mixing so you have to assume then that the deposition is homogeneous for the area you are sampling in (see the general comment on this above).
- Line 371: What do the grey filled circles correspond to? All the other stations samples in this study?
- Line 373: That an individual measurement is higher than the average, or that of a higher dust loading, is no surprise and does not require that there a rapid removal mechanism needs to be invoked as it depends on the time scale for the processes. In this case it the dust loading appears to be critical and as mentioned above there is an inverse relationship to the fractional solubility (Shelley et al., 2018).
- Line 391 and 405: Figure 6 (b). If this is a simple two component model the data would actually fit a curve if the value of Fe:Al was constant in the biogenic and lithogenic silicate fractions using a linear function as is computed here the intercept is not the actually the value of Fe:Al in the biogenic fraction and is in fact an underestimate of it. The authors can easily check this for themselves by plotting over the top values of the Fe/Al ratio at different LSi/BSi ratios assuming fixed Fe/Al for the end members. It also implies that the Fe:Al ratio for the LSi is less than the biogenic value which is what it should be for a bioactive element like Fe. Moral of the story is always be wary of linear fits to ratio plots!
- Line 405: Figure 6: It would be helpful for the reader if you also include the Fe/Al ratio for Saharan dust or continental crust.
- Line 405: Figure 6 what about the Fe/Al for stations ST04 and ST05 where the large excess of particulate aluminium is found, if this is indeed dust going down through the water column it should have a similar Fe/Al ratio as for the dust itself but the manuscript is completely silent on this! So please show the Fe/Al for these stations and the particulate iron profiles.
- Line 419: This is for dust inputs of 1.6 and 8.8 g m-2 for a single dust event! With almost no change in the dissolved iron concentrations, this seems rather remarkable.

- Line 445: How about a nepheloid layer, resuspension of material transported to the interior. In regard to the timescales, it is also important to consider the role of colloids here (see the general comment on this above).
- Line 456: Though this process is also part of the low mesopelagic Fe regeneration efficiency referred to on line 445 so there is a contradiction here if bacterial remineralization is being suggested here but not earlier.

**References:**

- Abbott, M.R., Brink, K.H., Booth, C.R., Blasco, D., Codispoti, L.A., Niiler, P.P., Ramp, S.R., 1990. Observations of Phytoplankton and Nutrients from a Lagrangian Drifter off Northern California. Journal of Geophysical Research 95, 9393-9409.
- Baker, A.R., Croot, P.L., 2010. Atmospheric and marine controls on aerosol iron solubility in seawater. Marine Chemistry 120, 4-13.
- Berger, C.J.M., Lippiatt, S.M., Lawrence, M.G., Bruland, K.W., 2008. Application of a chemical leach technique for estimating labile particulate aluminum, iron, and manganese in the Columbia River plume and coastal waters off Oregon and Washington. Journal of Geophysical Research: Oceans 113, C00B01.
- Bergquist, B.A., Wu, J., Boyle, E.A., 2007. Variability in oceanic dissolved iron is dominated by the colloidal fraction. Geochimica et Cosmochimica Acta 71, 2960-2974.
- Boyd, P.W., Law, C.S., Hutchins, D.A., Abraham, E.R., Croot, P., Ellwood, M.J., Frew, R.D., J, H., S, H., Hare, C.E., Higgins, J., Hill, P., Hunter, K.A., Leblanc, K., Maldonado, M., McKay, R.M.L., Oliver, M., Pickmere, S., Safi, K., Sanudo-Wilhelmy, S., Strzepek, R.F., Tovar-Sanchez, A., Wilhelm, S.W., 2005. FeCycle: Attempting an iron biogeochemical budget from a mesoscale SF6 tracer experiment in unperturbed low iron waters. Global Biogeochemical Cycles 19, GB4S20, 10.1029/2005GB002494.
- Caschetto, S., Wollast, R., 1979. Vertical distribution of dissolved aluminium in the mediterranean sea. Marine Chemistry 7, 141-155.
- Cutter, G.A., Andersson, P., Codispoti, L., Croot, P., Francois, R., Lohan, M., Obata, H., Rutgers van der Loeff, M., 2010. Sampling and Sample-handling Protocols for GEOTRACES Cruises. 2010 GEOTRACES Standards and Intercalibration Committee.
- Dammshäuser, A., Croot, P.L., 2012. Low colloidal associations of aluminium and titanium in surface waters of the tropical Atlantic. Geochimica et Cosmochimica Acta 96, 304-318.
- Dammshäuser, A., Wagener, T., Croot, P.L., 2011. Surface water dissolved aluminum and titanium: Tracers for specific time scales of dust deposition to the Atlantic? Geophys. Res. Lett. 38, L24601.
- Dammshäuser, A., Wagener, T., Garbe-Schönberg, D., Croot, P., 2013. Particulate and dissolved aluminum and titanium in the upper water column of the Atlantic Ocean. Deep Sea Research Part I: Oceanographic Research Papers 73, 127-139.
- Folger, D.W., Burckle, L.H., Heezen, B.C., 1967. Opal Phytoliths in a North Atlantic Dust Fall. Science 155, 1243-1244.
- Gdaniec, S., Roy-Barman, M., Foliot, L., Thil, F., Dapoigny, A., Burckel, P., Garcia-Orellana, J., Masqué, P., Mörth, C.-M., Andersson, P.S., 2018. Thorium and protactinium isotopes as tracers of marine particle fluxes and deep water circulation in the Mediterranean Sea. Marine Chemistry 199, 12-23.
- Guieu, C., D'Ortenzio, F., Dulac, F., Taillandier, V., Doglioli, A., Petrenko, A., Barrillon, S., Mallet, M., Nabat, P., Desboeufs, K., 2020. Introduction: Process studies at the air–sea interface after atmospheric deposition in the Mediterranean Sea – objectives and strategy of the PEACETIME oceanographic campaign (May–June 2017). Biogeosciences 17, 5563-5585.
- Jickells, T.D., Liss, P.S., Broadgate, W., Turner, S., Kettle, A.J., Read, J., Baker, J., Cardenas, L.M., Carse, F., Hamren-Larssen, M., Spokes, L., Steinke, M., Thompson, A., Watson, A., Archer, S.D., Bellerby, R.G.J., Law, C.S., Nightingale, P.D., Liddicoat, M.I., Widdicombe, C.E., Bowie, A., Gilpin, L.C., Moncoiffé, G., Savidge, G., Preston, T., Hadziabdic, P., Frost, T., Upstill-Goddard, R., Pedrós-Alió, C., Simó, R., Jackson, A., Allen, A., DeGrandpre, M.D., 2008. A Lagrangian biogeochemical study of an eddy in the Northeast Atlantic. Progress in Oceanography 76, 366-398.

- King, A.L., Buck, K.N., Barbeau, K.A., 2012. Quasi-Lagrangian drifter studies of iron speciation and cycling off Point Conception, California. Marine Chemistry 128–129, 1-12.
- Krom, M.D., Thingstad, T.F., Brenner, S., Carbo, P., Drakopoulos, P., Fileman, T.W., Flaten, G.A.F., Groom, S., Herut, B., Kitidis, V., Kress, N., Law, C.S., Liddicoat, M.I., Mantoura, R.F.C., Pasternak, A., Pitta, P., Polychronaki, T., Psarra, S., Rassoulzadegan, F., Skjoldal, E.F., Spyres, G., Tanaka, T., Tselepides, A., Wassmann, P., Wexels Riser, C., Woodward, E.M.S., Zodiatis, G., Zohary, T., 2005. Summary and overview of the CYCLOPS P addition Lagrangian experiment in the Eastern Mediterranean. Deep Sea Research Part II: Topical Studies in Oceanography 52, 3090-3108.
- Landing, W.M., Bruland, K.W., 1987. The contrasting biogeochemistry of iron and manganese in the Pacific Ocean. Geochimica et Cosmochimica Acta 51, 29-43.
- Law, C.S., Abraham, E.R., Woodward, E.M.S., Liddicoat, M.I., Fileman, T.W., Thingstad, T.F., Kitidis, V., Zohary, T., 2005. The fate of phosphate in an in situ Lagrangian addition experiment in the Eastern Mediterranean. Deep Sea Research Part II: Topical Studies in Oceanography 52, 2911-2927.
- Law, C.S., Martin, A.P., Liddicoat, M.I., Watson, A.J., Richards, K.J., Woodward, E.M.S., 2001. A Lagrangian SF6 tracer study of an anticyclonic eddy in the North Atlantic: patch evolution, vertical mixing and nutrient supply to the mixed layer. Deep-Sea Res. Part II-Top. Stud. Oceanogr. 48, 705-724.
- Li, F., Ren, J., Yan, L., Liu, S., Liu, C., Zhou, F., Zhang, J., 2013. The biogeochemical behavior of dissolved aluminum in the southern Yellow Sea: Influence of the spring phytoplankton bloom. Chinese Science Bulletin 58, 238-248.
- Misic, C., Castellano, M., Ruggieri, N., Harriague, A.C., 2008. Variations in ectoenzymatic hydrolytic activity in an oligotrophic environment (Southern Tyrrhenian Sea, W Mediterranean). Journal of Marine Systems 73, 123-137.
- Moore, R.M., Millward, G.E., 1984. Dissolved-particulate interactions of aluminium in ocean waters. Geochimica et Cosmochimica Acta 48, 235-241.
- Moran, S.B., Moore, R.M., 1988. Temporal variations in dissolved and particulate aluminium during a spring bloom. Estuarine, Coastal and Shelf Science 27, 205-215.
- Moran, S.B., Moore, R.M., 1989. The distribution of colloidal aluminum and organic carbon in coastal and open ocean waters off Nova Scotia. Geochimica et Cosmochimica Acta 53, 2519-2527.
- Mullin, J.B., Riley, J.P., 1955. The colorimetric determination of silicate with special reference to sea and natural waters. Analytica Chimica Acta 12, 162-176.
- Murphy, J., Riley, J.P., 1962. A modified single solution method for the determination of phosphate in natural waters. Analytica Chimica Acta 27, 31-36.
- Nelson, D.M., Smith, W.O., Muench, R.D., Gordon, L.I., Sullivan, C.W., Husby, D.M., 1989. Particulate matter and nutrient distributions in the ice-edge zone of the Weddell Sea: relationship to hydrography during late summer. Deep Sea Research Part A. Oceanographic Research Papers 36, 191-209.
- Paasche, E., 1980. Silicon content of five marine plankton diatom species measured with a rapid filter method1. Limnol. Oceanogr. 25, 474-480.
- Planquette, H., Fones, G.R., Statham, P.J., Morris, P.J., 2009. Origin of iron and aluminium in large particles (>53 μm) in the Crozet region, Southern Ocean. Marine Chemistry 115, 31-42.
- Ragueneau, O., Savoye, N., Del Amo, Y., Cotten, J., Tardiveau, B., Leynaert, A., 2005. A new method for the measurement of biogenic silica in suspended matter of coastal waters: using Si:Al ratios to correct for the mineral interference. Continental Shelf Research 25, 697-710.
- Rolison, J.M., Middag, R., Stirling, C.H., Rijkenberg, M.J.A., de Baar, H.J.W., 2015. Zonal distribution of dissolved aluminium in the Mediterranean Sea. Marine Chemistry 177, Part 1, 87-100.
- Roy-Barman, M., Coppola, L., Souhaut, M., 2002. Thorium isotopes in the western Mediterranean Sea: an insight into the marine particle dynamics. Earth and Planetary Science Letters 196, 161-174.

- Roy-Barman, M., Lemaître, C., Ayrault, S., Jeandel, C., Souhaut, M., Miquel, J.C., 2009. The influence of particle composition on Thorium scavenging in the Mediterranean Sea. Earth and Planetary Science Letters 286, 526-534.
- Schroeder, K., Cozzi, S., Belgacem, M., Borghini, M., Cantoni, C., Durante, S., Petrizzo, A., Poiana, A., Chiggiato, J., 2020. Along-Path Evolution of Biogeochemical and Carbonate System Properties in the Intermediate Water of the Western Mediterranean. Frontiers in Marine Science 7.
- Shelley, R.U., Landing, W.M., Ussher, S.J., Planquette, H., Sarthou, G., 2018. Regional trends in the fractional solubility of Fe and other metals from North Atlantic aerosols (GEOTRACES cruises GA01 and GA03) following a two-stage leach. Biogeosciences 15, 2271-2288.

---

## Author Comment (AC1)

**Responses to Referee #2 comments (manuscript #bg-2021-87)**

**Overview**

This manuscript presents data on dissolved and particulates phases for iron and aluminium collected in the Mediterranean during the PEACETIME project. During the expedition a number of dust/precipitation events occurred and the authors attempt to derive dust fluxes from measurements before and after such events. Unfortunately the are missing some before data and have used data from other stations to try to bridge this gap, without taking into account possible differences in hydrology and past depositional histories between the stations concerned, this results in what appears to be a major overestimation of the estimated dust fluxes at some stations due to the use of inappropriate initial conditions. The lack of a Lagrangian hydrographic context for the stations also suggests that horizontal variations are being interpreted as temporal changes in the vertical.

We thank the reviewer for these constructive comments. Please find a point-by-point response to these comments.

**General comments**

**General comment #1** – *Influence of Hydrography, Eulerian or Lagrangian approaches*: A critical omission at present from this work is that there is no information on the hydrography and what is going on in the water. For a process study this is a major flaw as all too often changes in water masses may be interpreted falsely as the effect of rapid in water column processing. This is the reason why Lagrangian experiments are carried out in the ocean (Abbott et al., 1990; Boyd et al., 2005; Jickells et al., 2008; King et al., 2012; Krom et al., 2005; Law et al., 2005; Law et al., 2001) in order to remove the effects of water mass movement as much as possible. While the optimal way to carry out such a study is to use a tracer such as SF6 (Law et al., 2001), it can also be as simple as putting in a drifting buoy as the sampling point (King et al., 2012). It appears from the description in the associated paper (Guieu et al., 2020) that the same eulerian point was occupied for each station and no attempt at a quasi Langranian study was made, though that work mentions that a Lagrangian adaptive sampling strategy was employed, but this apparently refers to using satellite data to avoid placing stations at fronts or in eddies.

Concerning the sampling strategy, we ask the reviewer to refer to the Introduction paper of the special issue (Guieu et al., 2020). Briefly, the strategy was indeed to avoid dynamic areas such as fronts and small-scale eddies. This approach was successfully used during several other cruises such as LATEX (Doglioli et al., 2013), KEOPS (d'Ovidio et al., 2015), OUTPACE (Moutin et al., 2017), and OSCAHR (Rousselet et al., 2019). Van Wambeke et al. (in revision) demonstrated that changes in water masses were minor during the occupation of the PEACETIME long stations, and that the sampling strategy adopted allowed assessing the impact of atmospheric nutrient deposition on biogeochemical processes and fluxes. Quasi-lagrangian studies cited above were performed over >10-day (Law et al., 2001, 2005; Boyd et al., 2005; Krom et al., 2005; King et al., 2012) to 24-day period (Jickells et al., 2008). This is a longer time period compared to the 4/5-day occupation period at our 3 long stations, meaning that lateral advection and diffusion in these low-dynamics areas could be neglected (relative to air-sea exchanges).

**General comment #2** – *Absence of a hydrographical context*: I am really surprised that the authors did not think to use salinity as a tracer of recent precipitation events as this would also allow an estimate of the amount of rain that has fallen if there is minimal horizontal advection of water masses.

According to the dilution law ($Dh/h = DS/S$), a 10 mm precipitation event (such as the one predicted by the ARPEGE model over the Tyrrhenian sea; Supp. Fig. 3d) would lead to a decrease in salinity of <0.02 in surface waters assuming a mixed-layer depth of 20 m and an initial salinity of 37.7 (DS = 37.7

* 0.01 / 20). Such a small variation in salinity would be measurable in the first meter of the water column using the underway system (see Figure below for the TYR long station), the first meter being usually not sampled with the CTD. However, it is difficult to deconvoluate the effects of precipitations and potential horizontal mixing. Furthermore, precipitations occurred few days before we arrived on site.

[Figure]

[Figure]

Figure – Time evolution of surface salinity measured with the underway system at the TYR long station.

It also appears that there is data for transmission and a LISST-Deep, both of which presumably would have picked up changes in the particles in the deep waters where the apparent excess particulate aluminium was located.

To pick up changes in particles, we would need pre-depositional vertical profile of particle concentration. Furthermore, we have shown in a previous study that the optical signature of dust particles in oligotrophic surface waters 'disappears' very quickly after a major dust event of 10 g m$^{-2}$ (about 2-3 days after deposition; Bressac et al. 2012).

Referencing data to density levels and examining the different water masses present (Schroeder et al., 2020) would help to resolve some of the issues around the lack of any hydrographical context in a future revision.

By using density levels, the vertical profiles are difficult to read. Instead salinity vertical profiles are now included in the Supp. Info material. These salinity profiles allow observing the different water masses, and in particular the chore of the LIW.

***General comment #3*** – *Dissolved Al removal dominant over release from dust:* Using the median profile as the initial starting point for looking at changes in the particulate concentrations has a number of apparent flaws that are not fully explained or investigated in the current manuscript.

We acknowledge that the approach used to estimate the dust deposition flux is not perfect, but appears to be the best way to provide an estimate of the dust flux in the absence of direct measurement. Uncertainties are clearly mentioned in the manuscript (other sources of pAl, the sampling method, the time lag between deposition and sampling, the vertical resolution; see section 3.3).

(i) There may be significant differences in the hydrography between the station used as the median profile and the experimental station.

Vertical profiles of salinity have been added to the supp. info material.

(ii) similarly the experimental station may include a transient nepheloid layer due to water mass differences.

See our response to comment #Line 256.

(iii) the time scales of the inferred changes being different, that is a seasonal or slow moving transient change is interpreted as an instantaneous response to a dust event.

Evidences were provided in the manuscript revealing that pAl excess was related to a recent dust event. First, the spatial coverage of Al$_{excess}$ is in excellent agreement with maps of dust wet deposition provided

by the different models. In addition, a continuous decrease in surface pAl concentrations was observed at TYR over ~72 h (Fig. 4). This decrease in surface pAl concentrations was accompanied by subsequent increases at depth, reflecting the recent and predominant supply of pAl from the atmosphere. And finally, sinking dust particles were collected in the sediment traps deployed at 200 and 1000 m depth.

In the present work section 3.3.1 at present does not give the reader a good idea of how the fluxes were calculated and what assumptions were made in the calculations. It appears that the difference between the median profile and the most recent profile is assumed to represent the total flux of particulate aluminium to the region in a single event, this seems very optimistic.

In the revised version of the manuscript, we now detail the calculation and assumptions made to estimate the dust deposition fluxes. We also acknowledge the possibility that past dust events may have contributed to the $Al_{excess}$ signal. Past depositional history in the studied area was investigated using satellite observations and model outputs (SKIRON model). This analysis revealed that two dust deposition events likely occurred mid- and late April (F. Dulac, pers. comm.). One dust deposition event likely occurred mid-April over the Ionian Sea, and hence did not contribute to the $Al_{excess}$ observed over the Tyrrhenian Sea. The second dust event likely occurred between the 26[th] and 28[th] of April over the Tyrrhenian Sea, that is about 19 to 26 days before occupation of the area. However, we are not able to provide an estimate of the dust flux associated with this event – and repeated vertical pAl profiles (Fig. 4) and downward pAl fluxes (Table 1) indicate that the dust input likely occurred only few days before our observations. We can thus reasonably assume that most of the $Al_{excess}$ signal originated from the dust event of the 11[th] of May.

***General comment #4*** – *Elevated particulate aluminium at depth but what about iron?* One of the main mysteries in this paper is if there is also a vast excess of particulate iron at stations ST04 and ST05 as there is for the particulate aluminium as this data is not provided (or at least I could not find it). It would be useful to include the particulate Fe data in Table 2 for instance as a check on the dust deposition estimates using Aluminium.

Sampling for particulate trace elements (including Fe) using the TMC rosette was only performed at the three long stations (TYR, ION and FAST). At the short stations (including ST04, ST05 and ST06), only particulate Al concentrations were obtained from an associated paper dealing with the distribution of particulate biogenic barium (Ba; Jacquet et al., in revision). Jacquet et al. analysed their samples only for Ba, Sr, Ca, and Al (the latter was used to estimate the lithogenic Ba fraction). To clarify which parameters are available at each station, the table below has been added to the supp. info material.

**Supp. Table 1** – Overview of the parameters presented in this study. Particulate Al concentrations were obtained from both the classical rosette at all the stations (Jacquet et al., in revision), and from the trace metal-clean rosette at the 3 long stations (TYR, ION and FAST). Particulate Fe concentrations were only obtained from the trace metal-clean rosette at the 3 long stations.

| Station | Lat (°N) | Lon (°E) | dAl | dFe | pAl | pFe | Reference |
|---|---|---|---|---|---|---|---|
| ST04 | 37.98 | 7.98 | × | × | | | this study |
| | | | | | × | | Jacquet et al. (in rev.) |
| ST05 | 38.95 | 11.02 | × | × | | | this study |
| | | | | | × | | Jacquet et al. (in rev.) |
| TYR | 39.34 | 12.59 | × | × | × | × | this study |
| | | | | | × | | Jacquet et al. (in rev.) |
| ST06 | 38.81 | 14.50 | × | × | | | this study |
| | | | | | × | | Jacquet et al. (in rev.) |
| ST07 | 36.66 | 18.20 | × | × | | | this study |
| ION | 35.49 | 19.80 | × | × | × | × | this study |
| | | | | | × | | Jacquet et al. (in rev.) |
| ST08 | 36.21 | 16.63 | × | × | | | this study |
| ST09 | 38.14 | 5.84 | × | × | | | this study |
| FAST | 37.95 | 2.91 | × | × | × | × | this study |
| | | | | | × | | Jacquet et al. (in rev.) |
| ST10 | 37.95 | 2.92 | × | × | | | this study |

***General comment #5*** – *Time scales*: Earlier work (Caschetto and Wollast, 1979) showed strong seasonal changes in dissolved aluminium concentrations, which would suggest that the residence time for dissolved aluminium is on the order of months and certainly less than a year for the Mediterranean. At present the manuscript basically interprets every change on the same timescale as between repeat measurements but care has to be taken when the sampling is not truly Lagrangian as you may confuse variability at depth with fast removal or dissolution processes.

We acknowledge that seasonal changes in dissolved Al can be important in the Mediterranean Sea, suggesting a low residence time for dAl. However, the timescale considered in our study is much sorter (days) – and our results obtained in the Tyrrhenian Sea highlight an absence of change in dAl rather than a change that could be due to variability at depth. At the FAST station, the timescale considered is even shorter (hours to days). At this timescale, it appears reasonable to attribute changes in Kd(Al) to physico-chemical processes.

What the data in this paper do inform about however is the effect of deposition in surface waters and reconciliation of the timescales involved that control the strong salinity relationship with dissolved aluminium (Rolison et al., 2015) as it fits well to a two end member situation with low Al Atlantic surface water and high Al Eastern Mediterranean waters, however this implies rapid mixing of surface waters in the Mediterranean to maintain the observed gradient. Thus it would be helpful to examine in the current work how horizontal mixing may dilute and transport dissolved aluminium after a dust event. I appreciate that in the current manuscript this aspect is downplayed so as to simplify flux calculations but it does appear that it is a critical aspect for maintaining dissolved aluminium concentrations in the Mediterranean.

Work on thorium isotopes in the Mediterranean (Roy-Barman et al., 2002; Roy-Barman et al., 2009) also suggests that the dissolution of continental particles stored in the margins needs to also be considered. This does not appear to have been examined in detail by subsequent researchers but could be explored here briefly. Similarly for the elevated particulate aluminium it would be useful to explore this with reference to earlier work on the pressure effects on aluminium speciation (Moore and Millward, 1984).

Dissolution of continental particles stores in margins should led to an anomaly in dAl at depth. Our results revealed a different trend: the absence of dAl anomaly over the whole water column.

Concerning the pressure effect on aluminium speciation, we clearly demonstrated that the excess pAl was directly related to a dust deposition event. Exploring other mechanisms would be too speculative.

**General comment #6** – *Role of colloids missing from this work:* A further obvious omission in this paper is any mention of the role of colloids for Fe (Bergquist et al., 2007) and Al (Dammshäuser and Croot, 2012; Moran and Moore, 1989) in the surface ocean. Given that colloidal Al could also be part of the measured particulate aluminium pool observed in this study it would also be of benefit to the reader if there was some discussion on the potential role of colloids in controlling the fractional solubility of Al in seawater (Dammshäuser and Croot, 2012; Dammshäuser et al., 2011; Dammshäuser et al., 2013; Moran and Moore, 1989).

Indeed, we deliberately ignored the potential role played by colloids in the manuscript since colloidal Fe and Al were not measured, but we fully agree with the reviewer, colloids can play a key role in transferring soluble Fe to the particulate phase. However, colloidal Al is neither presented, nor evoked, in two studies cited by the reviewer (Dammshäuser et al., 2011, 2013). In addition, Moran and Moore (1989) presented data obtained in coastal waters where the organic matter pool differed strongly from the one encountered at our studied sites. They showed through direct measurements that colloidal Al, mainly found in surface waters, was consistently <5% of the dissolved fraction. Only Dammshäuser and Croot (2012) presented colloidal Al data obtained in open waters (eastern tropical North Atlantic). The authors showed that the dissolved Al fraction was dominated by the soluble phase (<10 kDa) and colloidal association was <3.5%. They suggest that "*association with functional organic groups in the colloidal phase is unlikely for Al and Ti*", and they were not able to conclude whether the removal of Al in surface waters occurs predominantly via direct transfer of hydrolyzed species into the particulate fraction or via the rapid turnover of the colloidal phase. As a consequence, and given that colloidal Fe and Al were not measured during the cruise, we believe that it would be too speculative to discuss the role played by colloids in this study.

**General comment #7** – *Biogenic silica in Saharan Dust*: Saharan dust can contain a significant amount of biogenic silica (Folger et al., 1967), either as phytoliths or freshwater diatoms. So there is the possibility that the biogenic silica may not be formed in situ. Alternatively the biogenic silica may be overestimated during the analysis due to lithogenic dissolution (Ragueneau et al., 2005). This typically is estimated using dissolved Al as a tracer for the lithogenic fraction, which appears not to have been done in this case. This is an opportunity missed really as a sequential dissolution protocol looking at both the Si and Al would have provided strong evidence for Al weakly bound to the particles (i.e. that released with an acetic acid step as was done in other studies (Berger et al., 2008; Landing and Bruland, 1987; Planquette et al., 2009). At present it looks like the biogenic Si is overestimated.

Unfortunately, BSi associated with dust particles was not measured in this study. One sentence has been added to the manuscript in order to evoke this potential atmospheric source of BSi, as follows: "*… we cannot exclude the aeolian transport of BSi associated with dust particles (Folger et al., 1967; Barkley et al., 2021)*".

In addition, dissolved Al concentrations were not measured during the first leach. However, two corrections were applied to the BSi. First, BSi were corrected for dissolution that may occur within the brine by measuring the excess in $Si(OH)_4$. Second, BSi were corrected for the interference from lithogenic Si during the 5-h digestion step by monitoring on a regular basis the increase in dissolved silica (using a 200 µL aliquot of the leaching solution). The extrapolated intercept (time zero) of the linear part of the dissolution curve corresponds to the BSi content of the sample (DeMaster, 1981).

**General comment #8** – *Precipitation estimates:* There are satellite data available for precipitation https://gpm.nasa.gov/data - did the authors examine this dataset to examine the spatial and temporal scales of precipitation during the expedition? This would likely support the model runs used currently.

Various model reanalyses are now available (e.g., ERA5, WRF). These reanalyses do not show any precipitation in the area between the 10th and 13th of May. However, the half-hourly, 0.1°-resolution Global Precipitation Measurement (GPM) mission Integrated Multi-satellitE Retrievals for GPM (IMERG) final run images of the rainfall rate (product GPM_3IMERGHH_v06; Huffman et al., 2015) report the occurrence of light rains in the Tyrrhenian Sea in the early morning of the 11th of May, namely between 4 and 5:30 UTC, especially around 12°E and between 39°N and 40°N (<1 mm of accumulated precipitation, with a large error of up to several mm; Supp. Fig. 4 presented below). The study of the TRMM-3B42-v6 product, a former multi-satellite, 3-hourly precipitation product in the western Mediterranean region has shown that the detection of light rainfall is difficult when compared to rain gauge observations, with many occurrences missed by the satellite product (Sarrand et al., 2012). Indeed, the minimum corresponding random error image (Supp. Fig. 4, bottom) is 0.238 mm hr$^{-1}$ for pixels without detected rain, which can be assimilated to a lower detection limit. During the whole day, some light rain cells remain visible near Sardinia, Sicily or southern Italy. It is therefore well possible that a much larger area of the Tyrrhenian Sea than the one reported by the GPM IMERG images has been affected by light precipitation on that day. Unfortunately, we could not access Italian surface radar data that could better confirm the actual extension of rain.

**Supp. Fig. 4** – Top: Time averaged map of the GPM mission multi-satellite precipitation final run estimate with gauge calibration (mm hr$^{-1}$) over 04:00-05:30 UTC on the 11$^{th}$ of May 2017 in the western Mediterranean region, from the GPM_3IMERGHH v06, half-hourly, 0.1°-resolution. Bottom: Corresponding random error (mm hr$^{-1}$). After images produced by the Giovanni online data system.

[Figure]

There are also rain rate data included in the Guieu et al. (2020) work associated with this expedition.

Rain rate data included in Guieu et al. (2020) comes from the European radar composite products provided by the ODYSSEY system. However, the most coastal areas of the Tyrrhenian Sea, but not the central area of our TYR station, are monitored by Italian weather radars. Furthermore, up to now, Italian rain radar data have not been integrated to the ODYSSEE information system set-up in 2011 as part of the Operational Program for Exchange of Weather Radar Information (OPERA) (Huuskonen et al., 2014; https://www.eumetnet.eu/activities/observations-programme/current-activities/opera-radar-animation/), and we could not find access to an archive of these data.

***General comment #9** – Error estimates*: In the dust flux estimates there is no inclusion of error estimates and it would be useful to include these to constrain the values accordingly.

In the revised version of the manuscript, we now provide uncertainties for the dust flux estimates.

**Specific comments**

**Line 57**: In this context the work of Li et al. (2013) along with Moran and Moore (1988) are good examples of the apparent biological drawdown of Al.

These two references have been added.

**Line 68**: It is not that unexpected as the deposition of dust is known to scavenge other elements from the water column also.

The adjective 'unexpected' is used here to highlight the fact that dust deposition events have long been considered as a net source of dissolved iron for the surface ocean. Studies cited in this sentence were key in providing a more complex picture of the role of dust deposition in the ocean iron cycle.

**Line 69**: Do you mean fractional solubility with regard to aerosol solubility? As iron solubility in the water is a different measurement (Baker and Croot, 2010).

We meant iron solubility in seawater. This has been clarified.

**Line 94**: Please clarify that this is a Kevlar cable with a conducting insert in it and include information on the type of CTD that is on the rosette and the parameters measured on it.

This is now specified, as follows: "*At all stations, a 'classical' and a trace metal-clean (TMC) titanium rosette were deployed to sample the water column for biological and chemical parameters. Sensors present on these two rosettes and the parameters measured are detailed in Guieu et al. (2020). Samples for aluminium and iron analyses were collected using the TMC titanium rosette mounted with GO-FLO bottles deployed on a Kevlar cable with a dedicated clean winch, while samples for particulate Al (pAl) determination were also collected at all the stations from the classical rosette (see section 2.3; Supp. Table 1).*"

**Line 95**: The classical rosette is an aluminium oxide coated rosette? Please provide more information on this as the data in Supp Fig 2, do suggest that it is non-contaminating but the same data also suggest that the trace metal clean rosette could be contaminating (two obvious fliers). Alternatively it suggests there were major changes in the intermediate waters over a few hours (less likely).

The pAl vertical profile obtained with the TMC rosette at FAST (supp. Fig. 2b and e) is a composite profile. The 3 deepest samples (~1500-2500 m depth) were collected ~24 h after the samples collected within the 0-500 m depth range. The mismatch observed at 2000 and 2500 m depth could be explained by this time lag. At ION, different reasons can be invoked to explain the mismatch at 400 m depth (supp. Fig. 2a and d). First, it is very unlikely that the TMC rosette was contaminating – trace metal-clean protocol was strictly followed during the deployment, recovery, and sampling of the rosette – and particulate Zn, a contamination-prone trace element, did not reveal any contamination (not shown). In addition, nutrients concentrations were used to check that bottles were properly closed at the expected depth. Note that only concentrations of particulate Al, Ti and Fe were remarkably high at this depth. Therefore, it is likely that the presence of rare large lithogenic particles present at this depth horizon and collected only by 1 of our 2 sampling systems (i.e., the TMC rosette) led to this mismatch.

**Line 95**: Were nutrients taken from both sampling systems? There is no information here on if nutrients were taken and if so, how they were analysed.

Indeed, nutrients samples were collected from both sampling systems. These 2 datasets have been published by Van Wambeke et al. (in revision) and Pulido-Villena et al. (in revision). Since these parameters are not used in our study, they are not mentioned in the Materials and Methods section.

**Line 112**: There are no GEOTRACES consensus samples reported here for Al. Does this mean none were run? What data do you have that this method was giving good values?

The reviewer is correct that no GEOTRACES consensus sample were run for Al measurements. The method used for Al measurements in this study is adapted to the high dAl concentrations encountered in the Mediterranean Sea, but unadapted for measurements of concentrations lower than 10 nmol kg$^{-1}$. Most of the GEOTRACES consensus samples are in this low concentration range. For this reason, the precision of the measurements was assessed with regular measurements of an internal standard constituted of a filtered and acidified Mediterranean water. The concentration of this internal standard has been regularly measured during the cruise (53.5 ± 0.6 nmol kg$^{-1}$ (1 sd); n = 25). The accuracy relies on the intercomparaison of measurements performed during different Mediterranean cruises in the frame of the GEOTRACES IDP 2021.

**Line 113**: Are the data corrected for the reagent blank?

The dAl data were corrected for the reagent blank which represented systematically less than 3% of the measured values. This information has been added in the section 2.2.

**Line 120**: The GSC consensus value is 1.54 ± 0.12 nM for Fe and the data are available here: https://www4.obs-mip.fr/wp-content-omp/uploads/sites/31/2020/03/2019_Consensus_Values_2009_samples.pdf.

The GSC consensus value has been added.

**Line 125**: The GEOTRACES Cookbook is citable (Cutter et al., 2010).

This reference has been added.

**Line 140**: There are two or more fliers in Supp Fig 2, all of which come from the trace metal clean rosette – so what is the reason for this? Some are in the surface so this is understandable but at 400 m at ION and 2000 m FAST, this seems more like a bad bottle as it would not be expected that the water masses would change the much in the few hours between casts. Are there any other elements that can help with determining if it is contamination from a bottle?

See our response to comment #Line 95.

**Line 160**: The usual description for this is that it is a sequential leach using the method of Paasche (1980) as modified by Nelson et al. (1989) using the spectrophotometric method of Mullin and Riley (1955). I would avoid using the thesis as a citation as it quotes the wrong paper for the silicate method and while it has the authors correct for silicate, the journal reference is actually a paper on phosphate (Murphy and Riley, 1962).

We thank the reviewer for spotting this inappropriate citation. Methods published by Nelson et al. (1989), and Mullin and Riley (1955) are now cited.

**Line 165**: Table 1: The data for BSi, LSi, Al and Fe have high variability, with apparently one standard deviation being almost the same as the mean – more so in many cases than the variability in the mass fluxes. This is therefore an inherently noisy data set, which likely represents the challenge of short term deployments, so it is important that this is also discussed in the manuscript in more detail as concentrating on the average fluxes presents a somewhat distorted picture.

We are not sure to understand this remark. This temporal variability, also observed with long-term deployments (e.g., Miquel et al., 2011), is inherent to particle downward flux in the ocean. The average flux and the flux integrated over the whole duration of the deployment (4 to 5 days) are exactly the same, providing thus a real picture of the flux. By providing crucial information about the composition and magnitude of the exported particle flux, this dataset is key for interpreting our results.

**Line 172**: Was this always in the mixed layer as observed by the hydrographical measurements. See the general comment above regarding the absence of any hydrographic context in this work.

In order to provide an hydrographic context, the time evolution of the mixed-layer depth at the FAST station has been added to supp. Fig. 4 (see below).

[Figure]

[Figure]

Supp. Fig. 4 – Temporal and vertical resolutions of dissolved and particulate Fe and Al measurements performed at the FAST station within the (**a**) 0-20 m and (**b**) 0-200 m depth ranges. Grey-shaded areas indicate the two dusty rain events that occurred in the FAST station area. The grey-dotted vertical line corresponds to the time of the dusty rainfall sampled on board the R/V. The blue-dotted line corresponds to the depth of the mixed-layer. Note that only dissolved concentrations were measured the 03/06/17 (2nd profile).

**Line 185**: How was the mixed layer defined here? A density or temperature criteria or something else? More details should be supplied.

The mixed-layer depth (MLD) was shallow (~10-20 m), rapidly activated by mechanical effects of wind, and sampled at high frequency. As a consequence, an approach based on buoyancy criterion has been preferred to better resolve short term fluctuations in the mixing state, and the MLD was determined as the depth where the residual mass content is equal to 1 kg m$^{-2}$. Van Wambeke et al. (in revision), where the MLD calculation is detailed, is now cited in the manuscript.

**Line 219**: As mentioned in the general comments above, there is satellite data for precipitation so there is no need to invoke model predictions. The authors could also use the change in surface salinity as an indicator.

See our response to general comments #2 and #8.

**Line 239**: Can the authors provide a little more information on how this dust flux was determined – was it based on direct measurements of dust in the rain or via a proxy? This is important as without being able to see the Desboeufs et al. paper is may mean that this is actual a circular argument (e.g. that paper uses dissolved Al in the water to estimate the dust flux).

This dust flux was obtained by direct measurement of dissolved + particulate Al in rain samples. Please note that the Desboeufs et al. manuscript has been submitted (*Atmospheric Chemistry and Physics*), and will be available online soon.

**Line 252**: Using the median pAl profile without providing a hydrological context seems flawed in that it ignores differences in the hydrography. Additionally we are not provided with any statistical oversight of what the median profile represents. See the general comment on this above also.

To provide some statistical information of the medial pAl profile, 1st and 3rd quartiles have been added to Fig. 2a-d (grey-shaded areas). Also see our response to general comment #2.

[Figure]

**Line 256**: The high particulate concentrations at depth may also represent a nepheloid layer. Is there any transmission data from the CTD to examine this possibility? The Guieu et al. (2020) work indicates that there was a transmissometer and a LISST sensor onboard that could answer these questions so it would be a good idea to include this data here. I am aware of earlier work which has shown that this area can form nepheloid layers at intermediate depths (Misic et al., 2008). Nepheloid layers have also been shown to be important for scavenging of Th and Pa in the Mediterranean (Gdaniec et al., 2018).

Please see our response to general comments #2 and #3. In addition, Misic et al. (2008) reported nepheloid layers in coastal areas close to the Messina strait where the dynamic of water masses can create shear leading to the formation of transient nepheloid layers. Similarly, Gdaniec et al. (2018) reported benthic nepheloid layers. Our stations are considered as open sea stations and the pAl anomaly was not observed in the deeper layers of the water column, but in intermediate waters.

Alternatively it could be from a mesoscale eddy so it would be good to rule that also using sea surface altimetry.

Sea surface altimetry reveals that mesoscale eddies were rare in the Tyrrhenian Sea (see Guieu et al., 2020). Furthermore, these structures are usually present in the upper ~200 m, but to the best of our knowledge, rarely reach depths were the $Al_{excess}$ was observed.

**Line 267**: the units here are g m-2 but no time frame is given, this is then compared to an annual flux in g m-2 y-1 so how is the reader meant to compare this when we are not told how the deposition flux is calculated? The key missing piece of information is the assumed residence time for particulate aluminium and it seems here that the assumption (unstated) is that the excess particulate aluminium comes from a single dust event. See the general comments on this above.

See our response to general comment #3.

Line 270: This statement is confusing, as the inference appears to be that the excess dissolved Al all arrived in a single event? It is also not clear then how this relates to the rain event where the dust flux is estimated at 40 mg m-2.

A sentence has been added to clearly state that we assume that $Al_{excess}$ resulted from a single dust event, as follows: "*Assuming that Al represents 7.1% of the dust in mass (Guieu et al., 2002) and $Al_{excess}$ resulted from a single dust deposition event, a dust deposition flux ranging between 1.7 (ST06) and 8.9 g m$^{-2}$ (ST04) was derived from the $Al_{excess}$ inventory (Table 2). This range of dust deposition flux is of similar magnitude to the annual flux observed during former periods in the west central (7.4 g m$^{-2}$ yr$^{-1}$; Vincent et al., 2016) and northwestern Mediterranean Sea (11.4 g m$^{-2}$ yr$^{-1}$; Ternon et al., 2010), highlighting the remarkable magnitude of this event.*"

Concerning the wet dust deposition event observed at the FAST station, the following sentence was present in the manuscript: "*This sampled flux, considered as relatively modest compared to the multi-year record in this area (Vincent et al., 2016), ...*".

**Lines 277 & 281**: These two sentences are opposing each other and I don't follow here how if the deposition is assumed to be homogeneous that then the spatial variability indicates precipitation patchiness and not sampling bias. What type of sampling bias is inferred here?

We were referring to the time lag between deposition and observations. This has been clarified.

**Line 285**: Table 2 – deposition not 'depositon'

This has been corrected.

**Line 285**: Item 4: Why is the loss correction made? Particulate Aluminium in the traps is still in the water column at the time it is collected by the traps so it seems to be some sort of double accounting.

Once particulate Al reach 1000 m depth and is collected by the sediment trap deployed at this depth, it does not account for the 0-1000 m pAl inventory anymore (depth range used to estimate the dust deposition flux).

**Line 285**: Item 6: The authors should check these numbers as using 7.1% and a molecular weight for Al of 26.98 gives a dust flux of 9.18 g m-2 for the ST04 station when using 24169 µmol m-2 as the pAl input. Similarly 6.98 g m-2 for ST05.

We thank the reviewer for spotting these mistakes – Table 2 has been corrected accordingly.

**Line 289**: The vertical profiles for the particulate Fe data are not shown in this work, only the integrated values and the depth/time when sampled.

See our response to general comment #4.

**Line 298**: This is perhaps a good example of why a Lagrangian framework is better to use than a eulerian one as clearly different water masses with different deposition histories are passing through the same site so the interpretation here based on a eulerian approach is flawed. See the general comment on this above.

See our response to general comment #1.

**Line 300**: So what is the excess Al seen at ST04 and ST05, slow sinking dust particles? Why then are they not seen at FAST?

This is an interesting question. Unfortunately, we are not able to provide a clear answer to this question with our dataset.

**Line 305**: As Kd is a ratio, using a normalization approach as shown in Figure 3 makes it harder to follow. Furthermore the use of the 4th of June for the 0-200 m inventory makes comparison with the 0-20 m data even more problematic. Perhaps plotting the data as percentage of total Al might then be a better way to show how the relative amounts changed in the particulate and dissolved pools.

As proposed by the reviewer, we did try to plot the data as a percentage of total Al. However, since it did not improve the representation of the relative changes in particulate vs. dissolved pools, we chose to keep the Kd.

**Line 337**: See the general comment about the lack of details on the hydrography.

See our response to general comment #2.

**Line 340**: Please include error bars in Figure 4 as it is not clear what the variability of the measurements is relative to the apparent changes seen here.

Error bars have been added in Figure 4 (see below).

[Figure]

**Line 347**: Another good reference for this which includes data at high dust loadings is Shelley et al. (2018) and in that work it is clear that the fractional solubility is very low under high dust loadings. Though it may be higher when deposition is as rain.

This reference has been added.

**Line 359**: What is meant by '1D dynamic condition' here? That there is no lateral mixing? If the deposition is so patchy initially then of course lateral mixing will smooth this out quite quickly. Additionally horizontal mixing is typically orders of magnitude more than vertical mixing so you have to assume then that the deposition is homogeneous for the area you are sampling in (see the general comment on this above).

Indeed, '1D dynamic condition' is not appropriate since lateral (isotropic) mixing can happen. We refer to (1) the absence of lateral intrusion of different water masses, and (2) main physical mechanisms that could influence the dust dynamic were vertical (instead of horizontal).

**Line 371**: What do the grey filled circles correspond to? All the other stations samples in this study?

Grey filled circles correspond to data obtained during the PEACETIME cruise at the other short stations. This information has been added to the caption of Fig. 4.

**Line 373**: That an individual measurement is higher than the average, or that of a higher dust loading, is no surprise and does not require that there a rapid removal mechanism needs to be invoked as it depends on the time scale for the processes. In this case it the dust loading appears to be critical and as mentioned above there is an inverse relationship to the fractional solubility (Shelley et al., 2018).

This remark has been taken into account, as follows: "*An Al fractional solubility of 5% was measured in rainwater for dust aerosols collected at the FAST station (Desboeufs et al., 2021), i.e., well above the conservative value of 1.5% obtained for a dust flux of 10 g m$^{-2}$ (Wuttig et al., 2013) and used to estimate dAl inputs over the Tyrrhenian Sea. This confirms that dust loading partly controls the Al fractional solubility (e.g., Shelley et al., 2018).*"

**Line 391 and 405**: Figure 6 (b). If this is a simple two component model the data would actually fit a curve if the value of Fe:Al was constant in the biogenic and lithogenic silicate fractions – using a linear function as is computed here the intercept is not the actually the value of Fe:Al in the biogenic fraction and is in fact an underestimate of it. The authors can easily check this for themselves by plotting over the top values of the Fe/Al ratio at different LSi/BSi ratios assuming fixed Fe/Al for the end members. It also implies that the Fe:Al ratio for the LSi is less than the biogenic value which is what it should be for a bioactive element like Fe. Moral of the story is always be wary of linear fits to ratio plots!

We thank the reviewer for this comment. Indeed, Fig. 6b strongly suggests that the relationship is better represented by a two-component model, and a linear fit is not appropriate here. Fig. 6b has been modified accordingly, as follows:

[Figure]

[Figure]

Figure 6: (a) Box whisker plot of the Fe/Al molar ratio (mol mol$^{-1}$) for suspended particles collected at TYR, ION, and FAST. The Fe/Al median values are 0.27 ($n = 37$; 0-20 m), 0.24 ($n = 13$; 20-60 m), 0.21 ($n = 17$; 60-100 m), 0.24 ($n = 12$; 100-200 m), 0.24 ($n = 9$; 200-500 m), and 0.24 ($n = 10$; 500-3000 m). For clarity, two outliers (Fe/Al = 0.50 and 1.02) observed in the 0-20 m depth range are not represented. (b) Fe/Al versus LSi/BSi molar ratios (mol mol$^{-1}$) of sinking particles collected at ~200 m (circles), 500 m (squares), and 1000 m depth (triangles) at the stations TYR (red), ION (purple), and FAST (yellow). The two black-dotted curves correspond to simple two-component models with BSi having a fixed Fe/Al molar ratio of 0.21 mol mol$^{-1}$ (i.e., value observed in the DCM in (a)), and LSi having a fixed ratio of 0.26 (Desboeufs et al., 2021; lower curve) or 0.30 mol mol$^{-1}$ (Guieu et al., 2002; upper curve). The yellow-shaded areas in (a) and (b) represent the range in Fe/Al molar ratio proposed for Saharan dust, with the lower limit corresponding to dust aerosols collected during the cruise at FAST (0.26 mol mol$^{-1}$; Desboeufs et al., 2021) and the upper limit to Saharan dust end-member (0.30 mol mol$^{-1}$; Guieu et al., 2002).

**Line 405**: Figure 6: It would be helpful for the reader if you also include the Fe/Al ratio for Saharan dust or continental crust.

Fe/Al ratios for Saharan dust measured during the cruise by Desboeufs et al. (2021) and for Saharan dust end-member (Guieu et al., 2002) have been added to the Figure 6. See our response to the previous comment.

**Line 405**: Figure 6 – what about the Fe/Al for stations ST04 and ST05 where the large excess of particulate aluminium is found, if this is indeed dust going down through the water column it should have a similar Fe/Al ratio as for the dust itself but the manuscript is completely silent on this! So please show the Fe/Al for these stations and the particulate iron profiles.

See our response to general comment #4.

**Line 419**: This is for dust inputs of 1.6 and 8.8 g m-2 for a single dust event! With almost no change in the dissolved iron concentrations, this seems rather remarkable.

This absence of change in the surface mixed-layer is indeed remarkable, but in agreement with previous observations showing that major dust event (for instance 10 g m$^{-2}$) can led to a net decrease in dFe concentration of 0.7 nM in the first 15 m of the water column (Wagener et al., 2010).

**Line 445**: How about a nepheloid layer, resuspension of material transported to the interior. In regard to the timescales, it is also important to consider the role of colloids here (see the general comment on this above).

See our response to comment #Line 256.

**Line 456**: Though this process is also part of the low mesopelagic Fe regeneration efficiency referred to on line 445 so there is a contradiction here if bacterial remineralization is being suggested here but not earlier.

Low mesopelagic Fe regeneration efficiency does not necessary mean that the release in ligands is low since the net regeneration of Fe implies other processes such as scavenging, as revealed by Bressac et al. (2019) and Whitby et al. (2020).

**References**

Barkley, A. E., et al.: Atmospheric Transport of North African Dust-Bearing Supermicron Freshwater Diatoms to South America: Implications for Iron Transport to the Equatorial North Atlantic Ocean. Geophysical Research Letters, 48(5), e2020GL090476, 2021.

Boyd, P. W., et al.: FeCycle: Attempting an iron biogeochemical budget from a mesoscale SF6 tracer experiment in unperturbed low iron waters. Global Biogeochemical Cycles, 19(4), 2005.

Bressac, M., et al.: A mesocosm experiment coupled with optical measurements to assess the fate and sinking of atmospheric particles in clear oligotrophic waters. Geo-Marine Letters, 32(2), 153-164, 2012.

Dammshäuser, A., and Croot, P. L.: Low colloidal associations of aluminium and titanium in surface waters of the tropical Atlantic. Geochimica et Cosmochimica Acta, 96, 304-318, 2012.

Dammshäuser, A., et al.: Surface water dissolved aluminum and titanium: Tracers for specific time scales of dust deposition to the Atlantic?. Geophysical Research Letters, 38(24), 2011.

Dammshäuser, A., et al.: Particulate and dissolved aluminum and titanium in the upper water column of the Atlantic Ocean. Deep Sea Research Part I: Oceanographic Research Papers, 73, 127-139, 2013.

DeMaster, D. J.: The supply and accumulation of silica in the marine environment. Geochimica et Cosmochimica acta, 45(10), 1715-1732, 1981.

Desboeufs, K., et al.: Wet deposition in the remote western and central Mediterranean: A source of nutrients and trace metals for the marine biosphere?, Atmos. Chem. and Phys., submitted.

Doglioli, A. M., et al..: A software package and hardware tools for in situ experiments in a Lagrangian reference frame, J. Atmos. Ocean. Tech., 30, 1940–1950, https://doi.org/10.1175/JTECHD-12-00183.1, 2013.

d'Ovidio, F., et al.: The biogeochemical structuring role of horizontal stirring: Lagrangian perspectives on iron delivery downstream of the Kerguelen Plateau, Biogeosciences, 12, 5567–5581, https://doi.org/10.5194/bg-12-5567-2015, 2015.

Folger, D. W. et al.: Opal phytoliths in a North Atlantic dust fall. Science, 155(3767), 1243-1244, 1967.

Gdaniec, S., et al.: Thorium and protactinium isotopes as tracers of marine particle fluxes and deep water circulation in the Mediterranean Sea. Marine Chemistry, 199, 12-23, 2018.

Guieu, C., et al.: Chemical characterization of the Saharan dust end-member: Some biogeochemical implications for the western Mediterranean Sea. Journal of Geophysical Research: Atmospheres, 107(D15), ACH-5, 2002.

Guieu, C., et al.: Introduction: Process studies at the air–sea interface after atmospheric deposition in the Mediterranean Sea–objectives and strategy of the PEACETIME oceanographic campaign (May–June 2017). Biogeosciences, 17(22), 5563-5585, 2020.

Hersbach, H., et al.: ERA5 hourly data on single levels from 1979 to present. Copernicus Climate Change Service (C3S) Climate Data Store (CDS). DOI: 10.24381/cds.adbb2d47, 2018.

Huuskonen, A., et al.: The operational weather radar network in Europe, Bujll. Am. Meteorol. Soc., 95, 897–907, https://doi.org/10.1175/BAMS-D-12-00216.1, 2014.

Jacquet, S. H. M., et al.: Particulate biogenic barium tracer of mesopelagic carbon remineralization in the Mediterranean Sea (PEACETIME project), Biogeosciences Discuss., https://doi.org/10.5194/bg-2020-271, in review, 2020.

Jickells, T. D., et al.: A Lagrangian biogeochemical study of an eddy in the Northeast Atlantic. Progress in Oceanography, 76(3), 366-398, 2008.

King, A. L., et al.: A comparison of biogenic iron quotas during a diatom spring bloom using multiple approaches. Biogeosciences, 9(2), 667-687, 2012.

Krom, M. D., et al.: Summary and overview of the CYCLOPS P addition Lagrangian experiment in the Eastern Mediterranean. Deep Sea Research Part II: Topical Studies in Oceanography, 52(22-23), 3090-3108, 2005.

Law, C.S., et al.: A Lagrangian SF6 tracer study of an anticyclonic eddy in the North Atlantic: patch evolution, vertical mixing and nutrient supply to the mixed layer. Deep-Sea Res. Part II-Top. Stud. Oceanogr. 48, 705-724, 2001.

Law, C. S., et al.: The fate of phosphate in an in situ Lagrangian addition experiment in the Eastern Mediterranean. Deep Sea Research Part II: Topical Studies in Oceanography 52, 2911-2927, 2005.

Miquel, J. C., et al.: Dynamics of particle flux and carbon export in the northwestern Mediterranean Sea: A two decade time-series study at the DYFAMED site. Progress in oceanography, 91(4), 461-481, 2011.

Misic, C., et al.: Variations in ectoenzymatic hydrolytic activity in an oligotrophic environment (Southern Tyrrhenian Sea, W Mediterranean). Journal of Marine Systems, 73(1-2), 123-137, 2008.

Moran, S. B., and Moore, R. M.: The distribution of colloidal aluminum and organic carbon in coastal and open ocean waters off Nova Scotia. Geochimica et Cosmochimica Acta, 53(10), 2519-2527, 1989.

Moutin, T., et al.: Preface: The Oligotrophy to the UlTra-oligotrophy PACific Experiment (OUTPACE cruise, 18 February to 3 April 2015), Biogeosciences, 14, 3207–3220, https://doi.org/10.5194/bg-14-3207-2017, 2017.

Pulido-Villena, E., et al.: Phosphorus cycling in the upper waters of the Mediterranean Sea (Peacetime cruise): relative contribution of external and internal sources, Biogeosciences Discuss. [preprint], https://doi.org/10.5194/bg-2021-94, in review, 2021.

Rousselet, L., et al.: Vertical motions and their effects on a biogeochemical tracer in a cyclonic structure finely observed in the Ligurian Sea, J. Geophys. Res.-Oceans, 124, 3561–3574, https://doi.org/10.1029/2018JC014392, 2019.

Van Wambeke, F., et al.: Influence of atmospheric deposition on biogeochemical cycles in an oligotrophic ocean system, Biogeosciences Discuss. [preprint], https://doi.org/10.5194/bg-2020-411, in review, 2020.

Wagener, T., et al.: Effects of dust deposition on iron cycle in the surface Mediterranean Sea: results from a mesocosm seeding experiment. Biogeosciences, 7(11), 3769-3781, 2010.

---

## Author Comment (AC2)

**Responses to Thomas Holmes (manuscript #bg-2021-87)**

**General comments**

The manuscript entitled "Subsurface iron accumulation and rapid aluminium removal in the Mediterranean following African dust deposition" by Bressac et al. is an interesting and well-written paper teasing apart the cycling of Fe and Al in the Mediterranean Sea and challenging the use of Al as a tracer of dust inputs to this region. Using a quick response dissolved and particulate water sampling regime, the authors were able to take high resolution time-series (hours to days) observations of two wet-deposition events while at sea and thus capture previously unobserved mechanisms controlling the cycling and removal of Fe and Al. This high-resolution sampling revealed that dissolved Fe increased in the surface layer during and at around 6h after deposition but was quickly scavenged to background concentrations, which is attributable to the saturation of Fe binding ligands in the region. Using Fe/Al concentration ratios in suspended and sinking particles, the authors were further able to show that phytoplankton, especially diatoms, actively accumulate Al. However, comparing Al/Si ratio of particles with published values for diatoms, the authors demonstrate that adsorption of Al onto biogenic particles, rather than active uptake by diatoms is the main sink for dissolved Al in the region. As a climate change hotspot, understanding current biogeochemical cycling mechanisms in the Mediterranean is important for ensuring the accuracy of modelled climate change impacts in this region. The mechanisms discussed in this study will be important for both optimisation of regional climate models and for future biogeochemical studies focusing on nutrient cycling in the Mediterranean. In my opinion, the scientific method utilised in this study is robust and the paper seems quite polished. I have few suggestions on the scientific discussion itself and have mainly suggested minor technical corrections. As such, this manuscript will be well suited for publication in Biogeosciences after minor revisions.

We thank Thomas Holmes for his constructive comments. Please find a point-by-point response to these comments.

**Specific comments**

**Section 3**: You refer to Supp. Fig. 3 quite regularly in the results section, and it is quite an interesting figure. It might be worth moving this figure into the text.

The figure below has been added to the revised manuscript. Maps of precipitation and wet dust deposition provided by the models are in the supp. info material.

Figure – (**a**) MSG/SEVIRI-derived daily (daytime) mean aerosol optical depth; the white ellipse on the 11th of May image includes the location of the 5 Tyrrhenian stations reported in Fig. 1; (b) Left: Time averaged map of the GPM mission multi-satellite precipitation final run estimate with gauge calibration (mm hr$^{-1}$) over 04:00-05:30 UTC on the 11th of May 2017 in the western Mediterranean region, from the GPM_3IMERGHH v06, half-hourly, 0.1°-resolution (Huffmann et al., 2015). Right: Corresponding random error (mm hr$^{-1}$). After images produced by the Giovanni online data system (Acker and Lepkouth, 2007).

[Figure]

(**a**) MSG/SEVIRI AOD$_{550}$

(**b**) GPMM precipitation on 11 May 2017 (4:00-5:300 UTC), and standard error (mm hr$^{-1}$)

Can you comment at all on what impact future changing climate conditions might have on Mediterranean Fe/Al cycling in the context of your results? It seems that there is uncertainty as to whether dust deposition events are predicted to increase or decrease in the region based on observed trends over the last few decades (e.g. https://doi.org/10.1016/j.atmosenv.2020.117736), but though it may seem speculative I think it could still be valuable to mention the impacts that future changes may have on Fe/Al cycling based on your results.

A discussion about the impact of future climate conditions has been added to the revised manuscript. In particular, results obtained during minicosm experiments performed during the PEACETIME cruise, and investigating the impact of dust deposition on Mediterranean plankton communities under future conditions of pH and temperature (https://doi.org/10.5194/bg-2020-202; https://doi.org/10.5194/bg-18-2663-2021), are used to discuss the potential impacts that future changes could have on the Fe and Al cycles.

**Technical corrections**
Line 55: change to 'solubility as Fe'
Line 57: remove 'the' in 'the biological activity'
Line 64: change to either 'where dust is free' or 'where dust particles are free'
Line 65: change to 'they demonstrated'
Line 70: remove second comma and 'in fine'
Line 75: change 'under' to 'in'
Line 80: change 'contrasted' to 'contrasting'
Line 90: change 'investigate' to 'investigating'
Line 150: change 'weighted' to 'weighed'
Line 185: perhaps add a mean or range for the Chl a concentrations observed during the cruise.
Line 197: change 'could' to 'were'

Lines 203 – 208: this sentence is too long and needs to be broken up to make it easier to read. Suggest adding full stop after '10th of May' on line 205, removing the next 'with' and then adding 'was observed' after '…Dulac et al., 1992)' on line 207.

Line 210: change 'was' to 'were'

Line 221: replace 'that day' with '11th of May' and remove from end of sentence.

Line 222: does '…1.5 g m-2 over 6 h, or more in the area of…' mean that there was more than 1.5 g m-2 forecasted at just stations 5, TYR and 6, or more than 1.5 g m-2 forecasted over the whole area? If it's the latter, I would replace 'with up to' with 'with at least', and remove 'or more'.

Line 236: 'occurred on the 3rd'. Also, can you define 'neighbouring area'?

Line 238: 'R/V on the 5th…'

Line 239: 'clear dust signature' isn't technically wrong, and readers of this paper wouldn't think you meant that the dust was clear, but it's a little ambiguous, so I would recommend changing to '… by a dust signature clearly revealed by the chemical composition of the rain,…'

Line 246: 'as do all'

Line 279: 'This is an order-of-magnitude…'

Line 280: replace 'with' with 'compared to' and 'indicates' with 'indicating'

Table 2: in footnote 6, change 'a' to 'an'

Line 295: change 'At the opposite' to 'In contrast'

Line 315: change 'increased both' to 'both increased'

Line 316: change 'was' to 'were'

Line 320: either change 'concentration' to 'concentrations', or add 'a' before 'remarkably', 'was before 'observed' and ', which' after the bracket.

Line 378: change 'phytoplanktons' to 'phytoplankton'

Line 388: change 'phytoplanktons' to 'phytoplankton'

Line 394: insert 'the' after (2)

Line 426: remove 'a'

Line 440: change 'week' to 'weeks'

Line 448: add 'water' after 'surface'

Line 450: add 'the' before 'low'

Line 460: change 'with' to 'in'

Line 472: This sentence needs a little restructuring. I suggest: '…central Mediterranean, we observed two atmospheric wet deposition events while measuring Al and Fe water column distributions, providing…'

Line 474: This sentence could also do with a slight tweak. I suggest: 'The water-column Al inventories were successfully utilised to assess deposition fluxes, complementing atmospheric…'

Line 481: change 'and extended until' to 'extending to' and 'excess to dFe' to 'excess of dFe'

We think the referee for spotting all these spelling mistakes. All these proposed corrections have been taken into account in the revised manuscript.

---

## Author Response (AR1)

We thank Christine Klaas for her work and her comments. Please find a point-by-point response to these comments.

Dear author, while your responses do address the main issues raised by the reviewers, I do have a similar issue raised by reviewer #2 concerning your dust flux estimates for the central Mediterranean. In your response to the comment for line 445 on nepheloid layers you refer to the answer for the comment on line 256 where you actually do not address the issue of advective transport from continental margins. This should be done more robustly by discussing circulation, water masses and other potential proxies.

A new section has been added to the manuscript (Section 4.1). In this section, the formation of nepheloid layers and their advective transport are proposed as a potential mechanism that in addition to deposition could also contribute to the subsurface excess in pAl observed at ST04 and ST05. We also conclude that this mechanism can be excluded at stations TYR and ST06 considering the geographical position, the bathymetry, and the water masses circulation in this area. Note that we were not able to find a proxy for these potential nepheloid layers. Indeed, the vertical profiles of beam transmission (figure below) showed a slight anomaly between 200 and 500 m depth. However, this anomaly could also be due to the dust deposition event.

**4.1 Advective transport from continental margins in the central Mediterranean Sea**

*In the absence of direct atmospheric measurements, large uncertainties are associated with the estimates of the dust deposition flux over the Tyrrhenian Sea. These uncertainties are partly driven by potential additional sources of pAl, and in particular the resuspension of sediments and their advective transport from continental margins (e.g., Misic et al., 2008). The Strait of Sicily, characterized by high turbidity values (Gdaniec et al., 2018), represents a zone of formation for nepheloid layers. Levantine intermediate waters (LIW) can then act as a conveyor belt that accumulates and transports particles from the eastern to western basin of the Mediterranean Sea (Taillandier et al., 2020). Stations ST04 and ST05, located in the southwestern sector of the Tyrrhenian Sea (i.e., the branch of circulation between the Strait of Sicily and the Sardinian Channel), could be potentially impacted by particles driven by this mechanism, contributing to the excess in pAl observed at those stations. At the opposite, the central part of the Tyrrhenian Sea is characterized by low turbidity values relative to the rest of the Mediteranean Sea (Gdaniec et al., 2018). During PEACETIME, lateral advection was negligible (A. Doglioli, pers. comm., 2020) at stations TYR and ST06 precluding any contribution of lithogenic particles other than from dust atmospheric deposition. Evidences of a recent dust deposition event over the Tyrrhenian Sea, traced by the excess in pAl, are discussed in the next section.*

[Figure]

Further, based on the flux data at TYR you estimate a sinking speed of ~180m/d. With this value in mind the Al excess observed in ST04 to TYR might not have been due to the deposition event on the 11th of March. The data shown in Fig.4 further supports this. The data from FAST (line 336) also suggest fast sinking speeds. Could this also explain the lack of increase in the dAl pool described in section 4.2.1.?

This sinking velocity (SV) of 180 m d$^{-1}$ is likely an upper limit and corresponds to rare large dust particles. Indeed, dust deposition on the ocean surface consists mostly of particles of a few microns in

size. For instance, Tafuro et al. (2006) investigated Saharan dust particles properties over the central Mediterranean basin. They observed a dominant coarse mode peaking at 1.7-3 μm, and an average coarse mode centered at ~2.2 μm at Lampedusa.

Laurenceau-Cornec et al. (2019) demonstrated that the Stokes law can be adequately used for estimating SV for dust particles and dust-loaded aggregates. According to the Stokes law and assuming a density of 2.6 g cm$^{-3}$ for dust particles, a SV of 180 m d$^{-1}$ would correspond to dust particles with a diameter of about 55 μm. Long-range transport and deposition of such large dust particles have already been observed (e.g., Van Der Does et al., 2018), however, they only represent a minor fraction of the flux and submicron-sized dust particles were likely responsible for the Al excess observed at our Tyrrhenian stations.

It is also unclear to the reader if the Al excess estimated for the above-mentioned stations have been corrected by the vertical fluxes measured at TYR (it seems they are only based on water column inventories). What would be the estimates of dust input when the excess is corrected for vertical export? Would the estimates be realistic?

*Calculation of the dust deposition flux over the Tyrrhenian Sea has been detailed, as follows: "Dust deposition flux over the Tyrrhenian Sea was estimated from the Al$_{excess}$ inventories corresponding to the difference between the measured 0-1000 m pAl inventories and a background 0-1000 m pAl inventory. In the absence of pre-depositional observations and historic pAl data (to the best of our knowledge), the median pAl vertical profile obtained during the cruise at the other stations unimpacted by this event (grey bold line on Fig. 3a-d), similar or slightly higher than pAl data available for the open Mediterranean Sea (e.g., Sarthou and Jeandel, 2001), was used as a background level. The comparison between the measured pAl vertical profiles and this background level revealed a marked excess in pAl south of Sardinia (ST04) and in the southern Tyrrhenian (ST05, TYR and ST06; Fig. 3a-d and Table 2). This spatial extent is in good agreement with the maps of precipitation and dust wet deposition provided for the 11$^{th}$ of May by the ARPEGE, SKIRON, and NMMB/BSC models (Supp. Fig. 3). The obtained Al$_{excess}$ inventories were further corrected for the loss of pAl associated with the sinking flux using the pAl downward flux measured at 1000 m depth at the TYR station (assuming a constant flux over the 3 to 10-day period after deposition). Assuming that Al represents 7.1% of the dust in mass (Guieu et al., 2002), and further assuming that Al$_{excess}$ resulted from a single dust event, a dust deposition flux ranging between 1.7 (ST06) and 9.2 g m$^{-2}$ (ST04) was derived from these Al$_{excess}$ inventories (Table 2). Large uncertainties are associated with these dust flux estimates, partly due to potential additional sources of pAl that are suspected for fluxes derived from ST04 and ST05 but unlikely for TYR and ST06 (see Section 4.1). Nevertheless, the approach remains valuable to estimate the magnitude of this dust event."*

In short, I agree with reviewer #2 that the evidence in support for a dust deposition as the origin for the Al observations at ST04 to TYR is not quite as robust. Further, based on the above, it seems estimates of dust input based on Al excess are fraught with substantial uncertainties that should be clearly stated if not estimated.

*Unfortunately, we are not able to estimate the uncertainties associated with these dust flux estimates. However, they are clearly stated, as follows: (i) "... We acknowledge that this approach involves uncertainties, as do all the observational approaches employed so far to quantify deposition (Anderson et al., 2016). Caveats include (1) other sources of pAl, and (2) some uncertainties into the derived dust fluxes that could come from the sampling method (Twining et al., 2015a), the time lag between deposition and sampling favouring dispersion of dust by lateral mixing, and to a lesser extent, the limited vertical resolution below 500 m depth (Fig. 3a-d).", and (ii) "... Large uncertainties are associated with these dust flux estimates – partly due to potential additional sources of pAl (discussed in Section 4.1) – and they must be taken cautiously...".* In addition, a new section has been added to the manuscript to discuss other potential sources of pAl in the Tyrrhenian Sea, highlighting the issues existing in deriving fluxes at ST04 and ST05 but reinforcing the confidence concerning flux estimates at TYR and ST06.

Line 94-95: please provide the relevant details of the "classical" CTD in the text instead of referencing to Guieu et al. (2020).

Details of the classical CTD has been added, as follows: "*The 'classical' CTD continuously measured temperature, salinity, dissolved oxygen concentration, photosynthetically active radiation, beam transmission (at 650 nm), and the chlorophyll a fluorescence.*"

Line 239: please specify in the text that values are based on measurement of dissolved + particulate Al of rain samples collected on board.
This is now specified, as follows: "*From the total (dissolved + particulate) Al concentration measured in this rainwater sample, a dust flux of $65 \pm 18$ mg m$^{-2}$ was measured (Desboeufs et al., in revision).*"

Legend Fig. 3: replace "Time evolution" with "Temporal evolution".
This has been corrected.

**References**

Laurenceau-Cornec, E.C., et al. (2020). New guidelines for the application of Stokes' models to the sinking velocity of marine aggregates. Limnology and Oceanography, 65(6), 1264-1285.

Tafuro A.M., et al. (2006). Saharan dust particle properties over the central Mediterranean. Atmospheric Research, 81(1), 67-93.

Van Der Does, M., et al. (2018). The mysterious long-range transport of giant mineral dust particles. Science advances, 4(12).

---

## Author Response (AR2)

We thank the Co-Editor-in-Chief, Christine Klaas, for all her suggestions that clearly improve this manuscript.

The four technical corrections asked by the Co-Editor-in-Chief have all been taken into account.

Best regards,
Matthieu Bressac on behalf the co-authors.